# MULTISET-EQUIVARIANT SET PREDICTION WITH APPROXIMATE IMPLICIT DIFFERENTIATION

**Yan Zhang**[*1] **David W. Zhang**[*2]

**Simon Lacoste-Julien**[1,3,4] **Gertjan J. Burghouts**[5] **Cees G. M. Snoek**[2]

Samsung - SAIT AI Lab, Montreal[1] University of Amsterdam[2]
Mila, Université de Montreal[3] Canada CIFAR AI Chair[4] TNO[5]

## ABSTRACT

Most set prediction models in deep learning use set-equivariant operations, but they actually operate on multisets. We show that set-equivariant functions cannot represent certain functions on multisets, so we introduce the more appropriate notion of *multiset-equivariance*. We identify that the existing Deep Set Prediction Network (DSPN) can be multiset-equivariant *without being hindered by set-equivariance* and improve it with approximate implicit differentiation, allowing for better optimization while being faster and saving memory. In a range of toy experiments, we show that the perspective of multiset-equivariance is beneficial and that our changes to DSPN achieve better results in most cases. On CLEVR object property prediction, we substantially improve over the state-of-the-art Slot Attention from 8% to 77% in one of the strictest evaluation metrics because of the benefits made possible by implicit differentiation.

## 1 INTRODUCTION

An example of *set prediction* is detecting the objects in an image: we want a collection of objects as output, but there is no inherent order to them. Multisets and sets are natural representations for such data due to their orderless nature, with the difference between them being whether duplicate elements are allowed or not. In general, they are useful for representing collections of *discrete and composable elements*, such as agents in an environment, instruments in a piece of music, or atoms in a molecule.

The usual approach to representing a multiset or set of vectors in memory is to store it as a list in which the elements are placed in an arbitrary order. To still treat the list as if it were a multiset or set, recent set prediction models avoid being affected by this arbitrary order by using list-to-list functions that are enforced to be *permutation-equivariant* (Zaheer et al., 2017): changing the order of the input elements should change the order of the output elements in the same way.

Now, consider this function that takes a multiset (represented as a list) of two scalars as input and "pushes them apart".

$$\texttt{push\_apart}([a,b]) = \begin{cases} [a-1, b+1] & \text{if } a \leq b \\ [a+1, b-1] & \text{otherwise} \end{cases} \tag{1}$$

For example, $\texttt{push\_apart}([1,2]) = [0,3]$ and $\texttt{push\_apart}([2,1]) = [3,0]$. At first glance, it appears that this function is permutation-equivariant because swapping the inputs swaps the outputs. However, notice that $\texttt{push\_apart}([0,0]) = [-1,1]$. Swapping the inputs has no effect in this case, so the outputs remain unchanged, thus $\texttt{push\_apart}$ is not permutation-equivariant.

This is a bit strange – the function only violates the equivariance property for equal input elements, in which case the order of the equal input elements is irrelevant anyway. How can we reconcile this observation with the standard definition of permutation-equivariance? Is this a problem in practice when there are rarely equal elements, but instead similar elements? We aim to address these question in our paper and present two key ideas:

---

[*]Equal contribution

Table 1: Difference between equivariance properties in the context of multisets. Only when satisfying *exclusive multiset-equivariance* (this paper) is it enforced that order does not matter while also allowing equal input elements to be mapped to different output elements like in `push_apart`.

| $f$ is set-equivariant | $f$ is multiset-equivariant | $f([\boldsymbol{a}, \boldsymbol{b}]) = [\boldsymbol{c}, \boldsymbol{d}]$ $\Leftrightarrow f([\boldsymbol{b}, \boldsymbol{a}]) = [\boldsymbol{d}, \boldsymbol{c}]$ | $f([\boldsymbol{a}, \boldsymbol{a}]) = [\boldsymbol{c}, \boldsymbol{d}]$ |
|:---:|:---:|:---:|:---:|
| ✓ | ✓ | guaranteed ✓ | not possible ✗ |
| ✗ | ✗ | not guaranteed ✗ | possible ✓ |
| ✗ | ✓ | guaranteed ✓ | possible ✓ |

1. **Multiset-equivariance.** We begin Section 2 by making the important observation that many existing models for set prediction thought to operate on sets actually operate on multisets. We then show that these models are too restricted in the functions they can represent on multisets due to the limitation of being permutation-equivariant, which we call *set-equivariance*. Due to continuity, this is a problem even when there are no exact duplicate elements. Thus, we develop a new notion of *multiset*-equivariance, which we argue is a more suitable property for operations on multisets.

   To avoid the aforementioned limitation of set-equivariance, a crucial point is that it is possible to be multiset-equivariant without being set-equivariant, which we call *exclusive multiset-equivariance*. We identify and show that deep set prediction networks (DSPN) (Zhang et al., 2019; Huang et al., 2020) satisfy this property with a specific choice of set encoder. This makes it the only set prediction model we are aware of that can satisfy exclusive multiset-equivariance.

2. **Implicit DSPN.** Despite this beneficial property, DSPN is outperformed by the set-equivariant Slot Attention (Locatello et al., 2020), which motivates us to improve other aspects of DSPN. We propose *implicit DSPN* (iDSPN) in Section 3: a version of DSPN that uses approximate implicit differentiation. Implicit differentiation enables us to use better optimizers and more iterations at a constant memory cost and less computation time. The approximation makes this faster to run and easier to implement by avoiding the computation of a Hessian while still having the same benefits.

In our experiments in Section 4, we start by evaluating the first point by comparing exclusively multiset-equivariant models with both set-equivariant and non-equivariant models on synthetic data. The former completely fail on this simple toy task while the latter are significantly less sample-efficient. Next, we evaluate the second point by testing the modeling capacity on autoencoding random sets, where iDSPN performs similarly to DSPN at the same iteration count and much better for the same computational cost. Lastly, iDSPN significantly raises the bar on CLEVR object property prediction, outperforming the state-of-the-art Slot Attention (Locatello et al., 2020) by a large margin of 69 percentage points on the $AP_{0.125}$ metric while only training for 7% of the number of epochs.

## 2 LIMITATIONS OF SET-EQUIVARIANCE

In this section, we begin by identifying problems that limit the expressivity of a wide class of set prediction models satisfying a type of permutation-equivariance, *set-equivariance*. We argue that set-equivariance is too strict and instead propose a relaxation of set-equivariance that is more suitable for set prediction models: *multiset-equivariance*. We end by discussing how an existing method, deep set prediction networks (Zhang et al., 2019; Huang et al., 2020), is *multiset*-equivariant but not *set*-equivariant and the associated benefits thereof (Table 1).

### 2.1 PRELIMINARIES

Sets in the context of deep learning are typically unordered collections of real-valued vectors. They are represented in memory as $n \times d$ matrices with the $n$ set elements in an arbitrary order, each element being a $d$-dimensional vector. Since elements in the matrix representation are typically not restricted to be unique, the term *multiset* is more appropriate. To still treat these matrices as multisets, operations must not rely on the arbitrary order of the elements. The two main properties to accomplish this are permutation-*invariance* and permutation-*equivariance* (Zaheer et al., 2017). In short, when changing the order of the elements in the matrix, the output of a function should either not change at all (invariance) or change order similarly (equivariance).

## 2.2 SET-EQUIVARIANT MODELS

Recent set prediction models (Lee et al., 2019; Locatello et al., 2020; Carion et al., 2020; Kosiorek et al., 2020) make use of set-to-set (permutation-equivariant list-to-list) functions to refine an initial set $Y_0$, which is usually a randomly generated or learnable matrix. Permutation-equivariance is desirable when processing sets because it prevents a function from relying on the arbitrary order of the set in its matrix representation. Such functions can be easily composed to build larger models that remain equivariant, which fits well into deep learning architectures as building blocks.

However, without having any restrictions on duplicate elements, it is more accurate to think of set prediction models as using multiset-to-multiset functions. This difference subtly changes the appropriate notion of equivariance for sets versus multisets, which will become important for our discussion on their limitations.

**Set-equivariance.** First, we generalize the standard notion of permutation-equivariance for sets to *multiset inputs* and refer to it as *set-equivariance*. This directly matches the usual definition of permutation-equivariance in the context of *set inputs* in the literature. Thus, many models already satisfy the following definition, including Slot Attention (Locatello et al., 2020), Set Transformers (Lee et al., 2019), TSPN (Kosiorek et al., 2020), and the equivariant DeepSets (Zaheer et al., 2017).

**Definition 1** (Set-equivariance for multiset inputs). A function $f : \mathbb{R}^{n \times d_1} \to \mathbb{R}^{n \times d_2}$ with multiset input $X$ is set-equivariant iff for any permutation matrix $P$,

$$f(PX) = Pf(X) \tag{2}$$

A consequence is that elements in a multiset that are equal before a set-equivariant function must remain equal after applying the function, as shown by the following:

**Proposition 1.** *For all permutation matrices* $P_1, P_2$ *such that* $P_1X = P_2X$, *a set-equivariant function* $f$ *satisfies:*

$$P_1 f(X) = P_2 f(X) \tag{3}$$

*Proof.*

$$P_1 f(X) = f(P_1X) = f(P_2X) = P_2 f(X) \tag{4}$$

$\square$

$P_1X = P_2X$ can be satisfied by $P_1 \neq P_2$ when there are duplicate elements in $X$, so $P_1 f(X) = P_2 f(X)$ means that they must remain duplicates. This inability of separating equal elements demonstrates that the classical definition of permutation-equivariance is overly restrictive when directly applied to multisets as in Definition 1. Therefore, we seek a more appropriate definition of permutation-equivariance for multiset inputs.

**Multiset-equivariance.** We now consider a slightly different notion of permutation-equivariance that no longer forces equal elements to remain the same: *multiset-equivariance*. Recall the push_apart function from Equation 1, which we suggest should be a valid equivariant function for multisets, but cannot be represented by a set-equivariant function. Therefore, we propose that there should be a distinction between set-equivariance and *multiset*-equivariance, the latter corresponding to the appropriate equivariance property for the typical multiset-to-multiset mappings used in set prediction. The key difference of multiset-equivariance is that *the output order for equal elements in the input multiset can be arbitrary*. This leads us to the following definition:

**Definition 2** (Multiset-equivariance for multiset inputs). A function $f : \mathbb{R}^{n \times d_1} \to \mathbb{R}^{n \times d_2}$ with multiset input $X$ is multiset-equivariant iff for any permutation matrix $P_1$, there exists a permutation matrix $P_2$ such that $P_1X = P_2X$ and:

$$f(P_1X) = P_2 f(X) \tag{5}$$

Any function that is set-equivariant is also multiset-equivariant (since we can set $P_2 = P_1$), but the converse is not true.

**The problem with set-equivariance.** We argue that set-equivariance is *too strong* of a property for set prediction. As we have shown in Proposition 1, it prevents equal elements from being processed differently (we demonstrate this experimentally in Section 4.1). More importantly, when the model is

continuous[1], *impossibility to separate equal elements means difficulty in separating similar elements*. For example, in order to learn the push_apart function, a model has to decide which element is larger. This decision is fundamentally discontinuous, but most models in deep learning cannot represent discontinuous jumps. A continuous model must approximate the discontinuity, which means that close to the discontinuity (where elements are similar) the modeling error will likely be high. A function like push_apart can for example be relevant in turning a multiset of random 3d points that are close together (i.e., similar) into a 3d shape with points that are farther away (Luo & Hu, 2021). Difficulty with separating similar elements can therefore lead to bad downstream performance. Similar elements could for example arise from a random initialization of $Y_0$, nonlinearities throughout the model zeroing out (ReLU) or squashing elements (sigmoid and tanh), the model naturally tending towards equal elements (perhaps from oversmoothing as is often seen in graph neural networks (Li et al., 2018)), or simply from being part of the dataset used.

**Summary.**    It is important to make the distinction between set-equivariance – as has traditionally been considered in the literature – and multiset-equivariance. Deep learning typically operates on multisets rather than sets, so the less restrictive multiset-equivariance *without set-equivariance* should be the desirable property. Most existing set prediction models are continuous and set-equivariant, which limits their modeling capacity even if exact duplicates are never encountered. We refer to models that are multiset-equivariant but not set-equivariant as *exclusively multiset-equivariant*.

## 2.3    Deep Set Prediction Networks

Here, we highlight the only set prediction model we are aware of that *can* be exclusively multiset-equivariant: deep set prediction networks (DSPN) (Zhang et al., 2019; Huang et al., 2020). The core idea of DSPN is to use backpropagation through a permutation-invariant encoder $g$ (multiset-to-vector) to iteratively update a multiset $Y$. The objective is to find a $Y$ that minimizes the difference between its vector encoding and the input vector $z$ (this vector could for example come from an input image being encoded). This is expressed as the following optimization problem:

$$L(Y, z, \theta) = ||g(Y, \theta) - z||^2 \tag{6}$$

$$\mathrm{DSPN}(z) = \arg\min_Y L(Y, z, \theta) \tag{7}$$

Because $g$ is permutation-invariant, any ordering for the elements in $Y$ has the same value for $L$. In the forward pass of the model, the $\arg\min$ is approximated by running a fixed number of gradient descent steps. In the backward pass, the goal is to differentiate Equation 7 with respect to the input vector $z$ and the parameters $\theta$ of the encoder. To do this, Zhang et al. (2019) unroll the gradient descent applied in the forward pass and backpropagate through each gradient descent step.

**Equivariance of DSPN.**    We now discuss the particular form of equivariance that DSPN takes, which has not been done before in the context of multisets. The gradient of the permutation-invariant encoder $g$ with respect to the set input $Y$ is always multiset-equivariant, but depending on the encoder, it is not necessarily set-equivariant. Zhang et al. (2019) find that FSPool-based encoders (Zhang et al., 2020) achieved by far the best results among the encoders they have tried. With FSPool, *DSPN becomes exclusively multiset-equivariant* to its initialization $Y_0$. This is due to the use of numerical sorting in FSPool: the Jacobian of sorting is exclusively multiset-equivariant (Appendix A).

Note that DSPN is not always exclusively multiset-equivariant, but it depends on the choice of encoder. A DeepSets encoder (Zaheer et al., 2017) – which is based on sum pooling – has the same gradients for equal elements, which would make DSPN set-equivariant. It is specifically the use of the exclusively multiset-equivariant gradient of sorting that makes DSPN exclusively multiset-equivariant.

## 3    Implicit Deep Set Prediction Networks

Despite being exclusively multiset-equivariant, DSPN is outperformed by the set-equivariant Slot Attention (Locatello et al., 2020). We begin by highlighting some issues in DSPN that might be the cause of this, then propose *approximate implicit differentiation* as a solution.

---

[1]There is a subtle difference between continuity in the usual sense and continuity on multisets. While the discussion here uses the usual notion, we elaborate on the consequences of *multiset-continuity* in Appendix B.

**Problems in DSPN.** Increasing the number of optimization steps for solving Equation 7 generally results in a better solution (Zhang et al., 2019), but requires significantly more memory and computation time and can lead to training problems (Belanger et al., 2017). To be able to backpropagate through Equation 7, the activations of every intermediate gradient descent step have to be kept in memory. Each additional optimization step in the forward pass also requires backpropagating that step in the backward pass. These issues limit the number of iterations that are computationally feasible (DSPN uses only 10 steps), which can have a negative effect on the modeling capacity due to insufficient minimization of Equation 7. We aim to address these problems in the following.

**Implicit differentiation.** Implicit differentiation is a technique that allows efficient differentiation of certain types of optimization problems (Krantz & Parks, 2012; Blondel et al., 2021). We replace the normal backward pass of DSPN with implicit differentiation and call our method *implicit deep set prediction networks* (**iDSPN**). Instead of differentiating through the entire forward optimization, implicit differentiation allows us to only differentiate the (local) optimum obtained at the end of the forward optimization in Equation 7. The minimizer $Y^*(z, \theta)$ is implicitly defined by the input vector $z$ and the encoder parameters $\theta$ through the fact that the gradient of $L$ at an optimum is 0:

$$\nabla_Y L(Y^*, z, \theta) = 0 \tag{8}$$

The implicit function theorem allows us to differentiate Equation 8 and obtain the Jacobians $\partial Y^*(z, \theta)/\partial z$ and $\partial Y^*(z, \theta)/\partial \theta$, which is all we need in order for iDSPN to fit into an autodiff framework. Appendix D contains the full details on how this works.

In the implicit differentiation, we need to solve a linear system involving the Hessian $H = \partial(\nabla_Y L)/\partial Y$ evaluated at $Y^*$ (also explained in Appendix D). This can be problematic when $H$ is ill-conditioned or singular, so we regularize it with the identity matrix through $H' = H/\gamma + I$ with $\gamma > 0$ (Martens, 2010; Rajeswaran et al., 2019). In the limit of $\gamma \to \infty$, the Hessian is simply replaced by the identity matrix. The limit can be interpreted as pretending that the inexact optimization of $L$ found an exact (local) minimum and that $L$ curves up equally in all directions around that point. Setting the Hessian equal to the identity matrix results in the following Jacobians, which are equivalent to backpropagating through a single gradient descent step at the optimum:

$$\frac{\partial Y^*(z, \theta)}{\partial z} = -\frac{\partial \left(\nabla_Y L(Y^*, z, \theta)\right)}{\partial z} \qquad \frac{\partial Y^*(z, \theta)}{\partial \theta} = -\frac{\partial \left(\nabla_Y L(Y^*, z, \theta)\right)}{\partial \theta} \tag{9}$$

Other motivations for this approximation are discussed in Fung et al. (2022), where they call this Jacobian-free backpropagation. Matching their results, we find that this is not only faster to run and easier to implement (no need to solve a linear system involving $H$), but also leads to better results than standard implicit differentiation in preliminary experiments. Hence, we apply this approximation to differentiate Equation 7 with respect to $z$ and $\theta$ in all of our experiments.

**Benefits.** Implicit differentiation saves a significant amount of memory in training because memory no longer scales with the number of iterations; only the size of the set and its optimizer determine the memory cost since updates to the set can be performed in-place. Because we only take the derivative at the final set, we decrease the computational cost by a factor of around $\frac{T-1}{2T}$, reducing it by nearly a half. The combined improvements mean that we can minimize the objective in Equation 7 better by increasing the number of optimization steps and using more sophisticated optimizers in the inner loop. The optimization procedure is treated as a black box now, so it does not even need to be differentiable.

**Further improvements.** A simple improvement enabled by implicit differentiation is the use of Nesterov's Accelerated Gradient for estimating $Y^*$. This has a better convergence rate on convex problems than standard gradient descent (Nesterov, 1983) and has found wide applicability in deep learning (Sutskever et al., 2013). Since the computation graph of the optimization is no longer kept in memory, it only has a minor impact on memory and computation: the momentum buffer is the same size as $Y$, updates can be performed in-place, and the additional momentum calculation does not need to be differentiated. We have also tried gradient descent with line search, but found that the slowdown due to additional evaluations of $L$ was not worth the small improvement in performance.

By default, we sample a random initial set $Y_0 \sim \mathcal{N}(0, I/10)$ to start the optimization with. Similar to DSPN, we can also use a learned initial set $Y_0$ to start closer to a solution. However, implicit differentiation treats the optimizer of Equation 7 as a black box, so there is no gradient signal for $Y_0$.

We therefore need to include a regularizer in Equation 6 to give us a gradient for $\boldsymbol{Y}_0$, for example by adding the regularizer from Rajeswaran et al. (2019): $\frac{\lambda}{2}||\boldsymbol{Y} - \boldsymbol{Y}_0||^2$. We use this only in Section 4.2 to make the forward passes of iDSPN and DSPN the same so that we can compare them fairly.

**Incorporating constraints.** We would like to have the ability to impose constraints on the outputs of iDSPN. For example, when each set element corresponds to a class label, we would like to enforce that every vector is in the probability simplex. As suggested by Blondel et al. (2021), we can incorporate such constraints through constrained optimization in implicit differentiation by changing Equation 8 to the following and then following the standard derivation in Appendix D:

$$\texttt{proj}(\boldsymbol{Y}^* - \nabla_{\boldsymbol{Y}} L(\boldsymbol{Y}^*, \boldsymbol{z}, \boldsymbol{\theta})) - \boldsymbol{Y}^* = \boldsymbol{0} \tag{10}$$

$\texttt{proj}$ is a function that projects its input onto the feasible region. We can then use projected gradient descent with this projection to ensure that the constraints are always satisfied. For example, $\texttt{proj}$ could map each element onto the closest point in the probability simplex, which is what we use in Section 4.1. Note that projections can map different elements to the same value, which can cause problems when $\nabla_{\boldsymbol{Y}} L$ is set-equivariant. By using an exclusively multiset-equivariant $\nabla_{\boldsymbol{Y}} L$, we avoid this problem because equal elements after a projection can be separated again.

When we already know parts of the desired $\boldsymbol{Y}$ (e.g. specific elements in the set or the values along certain dimensions), we can help the model by keeping those parts fixed and not optimizing them in Equation 7. For example, in Section 4.1 we know that the first few dimensions in the output multiset should be the exactly same as in the input multiset. If we fix these dimensions during optimization, the model only needs to learn the remaining dimensions that we do not already know. We use this in Section 4.1 (but not the other experiments) to make iDSPN exclusively multiset-equivariant to the input multiset, not just to the initialization $\boldsymbol{Y}_0$.

# 4 EXPERIMENTS

In this section, we evaluate three different aspects of our contributions: the usefulness of exclusive multiset-equivariance (Section 4.1), the differences between automatic and our approximate implicit differentiation (Section 4.2), and the applicability of iDSPN to a larger-scale dataset (Section 4.3). Appendix E contains additional experiments on these datasets, including further evaluation of exclusive multiset-equivariance. We provide detailed descriptions of the experimental procedure in Appendix F, show example inputs and outputs in Appendix G, and open-source the code to reproduce all experiments at `https://github.com/davzha/multiset-equivariance` and in the supplementary material.

## 4.1 DIFFERENCE BETWEEN EQUIVARIANCES ON CLASS-SPECIFIC NUMBERING

First, we propose a task that highlights the difference between models that are set-equivariant, exclusively multiset-equivariant, and not multiset-equivariant at all. Given a multiset like $[a, a, b, b, b]$, the aim is to number the elements for each class separately: $[(a, 0), (a, 1), (b, 0), (b, 1), (b, 2)]$. Since these are multisets, the numbered elements do not need to be in any particular order. Equal elements in the input are mapped to different elements in the output, so this is a task where being able to separate equal elements is directly necessary.

**Setup.** Each element in the input set of size 64 is generated by randomly picking a class out of 4 choices and one-hot encoding it. We have also considered different set sizes and number of classes, but varying these only affected the number of training steps and model parameters required for similar results. For the target output, a one-hot incremental ID that is separate for each class is concatenated to each element. A multiset matching loss is computed for all models (see details in Appendix F). We evaluate the accuracy of predicting every element in the multiset correctly.

**Results.** Table 2 shows our results. The two exclusively multiset-equivariant models DSPN and iDSPN perform well at any training set size. As expected, the set-equivariant models are unable to solve this task because they cannot map the equal elements in the input to different elements in the output. This applies even to transformers with random "position" embeddings, which similarly to TSPN (Kosiorek et al., 2020) and Slot Attention (Locatello et al., 2020) use noise to make elements

Table 2: Sample efficiency of equivariance properties on a task that requires separating equal elements. Accuracy in % (mean ± standard deviation) over 6 seeds, higher is better. PE refers to positional encoding, random PE refers to randomly sampled elements like in TSPN (Kosiorek et al., 2020).

| Model | Set-equivariant | Multiset-equivariant | Training samples | | |
|---|---|---|---|---|---|
| | | | $1\times$ | $10\times$ | $100\times$ |
| DeepSets | ✓ | ✓ | $0.0_{\pm 0}$ | $0.0_{\pm 0}$ | $0.0_{\pm 0}$ |
| Transformer without PE | ✓ | ✓ | $0.0_{\pm 0}$ | $0.0_{\pm 0}$ | $0.0_{\pm 0}$ |
| Transformer with random PE | ✗ | ✗ | $28.5_{\pm 8.1}$ | $22.6_{\pm 5.2}$ | $22.7_{\pm 7.7}$ |
| Transformer with PE | ✗ | ✗ | $0.0_{\pm 0}$ | $46.4_{\pm 26.2}$ | $94.4_{\pm 1.8}$ |
| BiLSTM | ✗ | ✗ | $0.0_{\pm 0}$ | $25.1_{\pm 20.2}$ | $\mathbf{97.9_{\pm 2.1}}$ |
| DSPN | ✗ | ✓ | $89.4_{\pm 2.4}$ | $88.9_{\pm 4.9}$ | $90.8_{\pm 4.2}$ |
| iDSPN (ours) | ✗ | ✓ | $\mathbf{96.8_{\pm 0.6}}$ | $\mathbf{98.1_{\pm 0.3}}$ | $97.9_{\pm 0.6}$ |

not exactly equal. Meanwhile, BiLSTM and transformers with normal positional encoding require at least $100\times$ more training samples to come close to the performance of iDSPN. This is because they lack the correct structural bias of not relying on the order of the elements, which makes them less sample-efficient. Appendix E.1 shows the effect of data augmentation for the non-equivariant models.

## 4.2 AUTOMATIC VERSUS IMPLICIT DIFFERENTIATION ON RANDOM MULTISETS

Next, we aim to evaluate our contributions in Section 3. To do this, we compare: 1) baseline DSPN, 2) iDSPN without momentum and 3) iDSPN with Nesterov's Accelerated Gradient. Comparing 1 and 2 allows us to see the difference between backpropagation through the optimizer and the approximate implicit differentiation (Equation 9) when using the same forward pass, while comparing 2 and 3 allows us to see the effects of using a better optimizer for the inner optimization. In Appendix E.2 we instead evaluate the difference between 1 (exclusively multiset-equivariant) and a modification to 1 where the FSPool is replaced with sum or mean pooling (set-equivariant).

We propose the task of autoencoding random multisets: every element is sampled i.i.d. from $\mathcal{N}(\mathbf{0}, \mathbf{I})$. The goal is to reconstruct the input with a permutation-invariant latent vector as bottleneck. Varying the set size $n$ and dimensionality of set elements $d$ allows us to control the difficulty of the task. Note that while this appears like a toy task, it is also likely harder than many real-world datasets with similar $n$ and $d$. This is because the independent nature of the elements means that there is no global structure for the model to rely on; the only way to solve this task is to memorize every element exactly, which is especially difficult when we want the latent space to be permutation-invariant. This task is also challenging because it indirectly requires a version of push_apart: similar elements in the initialization $\mathbf{Y}_0$ might need to be moved far away from each other in the output.

**Setup.** All models use the same encoder, so DSPN and iDSPN without momentum have exactly the same forward pass. We train each model on every combination of number of set elements $n \in \{2, 4, 8, 16, 32\}$, set element dimensionality $d \in \{2, 4, 8, 16, 32\}$, and optimization steps $\in \{10, 20, 40\}$. We evaluate the final validation reconstruction losses over 8 random seeds.

**Results.** In Figure 1, we show a selection of representative results. We show the full results in Appendix E.2 due to the large number of dataset and model combinations. When comparing the models at the same number of iterations (Figure 1, left), iDSPN performs similarly to DSPN, but

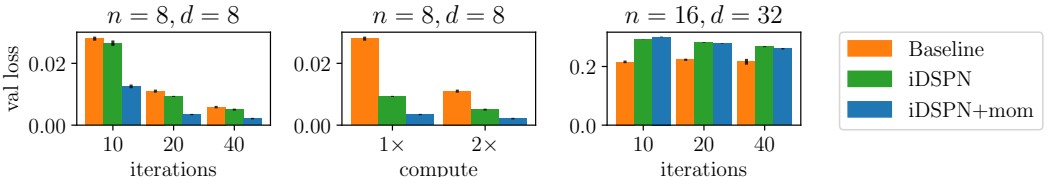

Figure 1: Implicit versus automatic differentiation for different dataset difficulties, reconstruction losses, lower is better. `iDSPN+mom` stands for iDSPN with Nesterov momentum. (left) and (right) compare models at equal iterations, (middle) compares models at roughly the same amount of computation. We select (right) as an atypical example where DSPN outperforms iDSPN.

iDSPN + momentum is a major improvement. This shows that the iDSPN gradients are of similar quality despite only using the approximate implicit differentiation, and that the improved optimizer leads to consistently better results. If we instead compare the models at roughly the same amount of computation (Figure 1, middle), it is clear that iDSPN has significant practical benefits over DSPN regardless of momentum. These results apply to most dataset configurations (see Appendix E.2).

The only settings where iDSPN + momentum performed worse than DSPN is when $n \times d$ approaches the dimensionality of the latent vector, 512 (Figure 1, right). In these cases, the set *encoder* runs into fundamental representation limits of set functions (Wagstaff et al., 2019). With performance being limited by the encoder, it is natural that better optimization yields little to no benefit. In Appendix G, it becomes clear that no model performs acceptably on these dataset configurations. When comparing the exclusively multiset-equivariant iDSPN to the set-equivariant iDSPNs in Appendix E.2, the former is consistently better, often with a lower loss by a factor of >10.

## 4.3 CLEVR OBJECT PROPERTY PREDICTION

Finally, we evaluate iDSPN on the CLEVR (Johnson et al., 2017) object property prediction task that was used in DSPN (Zhang et al., 2019) and Slot Attention (Locatello et al., 2020). Given an image of a synthetic 3d scene containing up to ten objects, the goal is to predict the set of their properties: 3d coordinate, size, color, and material (18 dimensions in total). To account for different set sizes in the dataset, a 19th dimension is added as a binary indicator for whether an element is present or not. Appendix E.3 compares an exclusively multiset-equivariant iDSPN to set-equivariant iDSPNs and contains results for DSPN in a like-to-like setting to iDSPN.

**Setup.** We mostly follow the experimental setup of DSPN where a ResNet (He et al., 2016) encodes the image into the input vector $z$, which is decoded into a set by iDSPN. We also include a few improvements that we elaborate on in Appendix F. Most notably, we use Nesterov's Accelerated Gradient as iDSPN optimizer and increase the number of optimization steps. The evaluation metric is a modified average precision ($AP_{threshold}$) where a predicted object is considered correct if every property is correct and the 3d coordinates are within the given Euclidean distance threshold of the ground-truth element.

**Results.** Table 3 shows our results. iDSPN significantly improves over any configuration of Slot Attention (see caption for description on the configurations * and †) on all AP metrics. It is better at attribute classification when ignoring the 3d coordinates ($AP_\infty$, 96.4% → 98.8%) and improves especially for the metrics with stricter 3d coordinate thresholds ($AP_{0.125}$, 7.9% → 76.9%). This is despite Slot Attention† using a three times higher weight on the loss for the coordinates than iDSPN. Note that a stricter AP threshold is always upper bounded by a looser AP threshold, so iDSPN is guaranteed to be better than Slot Attention† on $AP_{0.0625}$. Again, we see in Appendix E.3 that the exclusively multiset-equivariant iDSPN is better than the set-equivariant iDSPNs. Interestingly, iDSPN performs better than DSPN in a like-to-like setting (Appendix E.3) with the same number of iterations and same optimizer. This motivates thinking of iDSPN not as just an approximation of DSPN, but as a separate model altogether.

Table 3: Performance on CLEVR object property set prediction task, average precision (AP) in % (mean $\pm$ standard deviation) over 8 random seeds for our results, higher is better. Slot MLP and Slot Attention results are from Locatello et al. (2020), DSPN results are from Zhang et al. (2019). DSPN is trained at 10 iterations and evaluated at 30 iterations. For Slot Attention* and †, Locatello et al. (2020) supervise the model at each iteration and scale up the loss on the coordinates by either 2 or 6. We do not mark the iDSPN 256x256 image results in bold because of the difference in dataset setup.

| Model | Image size | $AP_\infty$ | $AP_1$ | $AP_{0.5}$ | $AP_{0.25}$ | $AP_{0.125}$ | $AP_{0.0625}$ |
|---|---|---|---|---|---|---|---|
| Slot MLP | 128x128 | $19.8_{\pm1.6}$ | $1.4_{\pm0.3}$ | $0.3_{\pm0.2}$ | $0.0_{\pm0.0}$ | $0.0_{\pm0.0}$ | — |
| DSPN | 128x128 | $85.2_{\pm4.8}$ | $81.1_{\pm5.2}$ | $47.4_{\pm17.6}$ | $10.8_{\pm9.0}$ | $0.6_{\pm0.7}$ | — |
| Slot Attention | 128x128 | $94.3_{\pm1.1}$ | $86.7_{\pm1.4}$ | $56.0_{\pm3.6}$ | $10.8_{\pm1.7}$ | $0.9_{\pm0.2}$ | — |
| Slot Attention* | 128x128 | $96.4_{\pm0.5}$ | $93.6_{\pm0.8}$ | $80.4_{\pm2.1}$ | $26.5_{\pm2.7}$ | $2.6_{\pm0.3}$ | — |
| Slot Attention† | 128x128 | $90.6_{\pm1.8}$ | $89.1_{\pm2.1}$ | $84.4_{\pm2.5}$ | $50.3_{\pm3.8}$ | $7.9_{\pm1.0}$ | — |
| iDSPN | 128x128 | $\mathbf{98.8}_{\pm\mathbf{0.5}}$ | $\mathbf{98.5}_{\pm\mathbf{0.6}}$ | $\mathbf{98.2}_{\pm\mathbf{0.6}}$ | $\mathbf{95.8}_{\pm\mathbf{0.7}}$ | $\mathbf{76.9}_{\pm\mathbf{2.5}}$ | $\mathbf{32.3}_{\pm\mathbf{3.9}}$ |
| iDSPN | 256x256 | $99.4_{\pm0.3}$ | $99.2_{\pm0.4}$ | $99.0_{\pm0.5}$ | $97.8_{\pm0.7}$ | $86.9_{\pm1.0}$ | $47.2_{\pm3.8}$ |

**Efficiency.** iDSPN also compares favorably against Slot Attention when considering training times. Locatello et al. (2020) train Slot Attention for 200k steps with batch size 512 while we only train for 100 epochs with batch size 128, which is equivalent to 55k steps. Their model takes around 51 hours to train on a V100 GPU, while our model only takes 2 hours and 40 minutes on a V100 GPU (though with different infrastructure around it). Ignoring possible speed-ups from increasing the batch size for our model, the per-sample iteration speed is similar.

## 5    RELATED WORK

Set prediction is ubiquitous due to the versatility of representing data as sets. Example applications include speaker diarization (Fujita et al., 2019), vector graphics generation (Carlier et al., 2020), object detection (Carion et al., 2020) and point cloud generation (Achlioptas et al., 2018). Since sets are closely related to graphs, there are also natural extensions to molecule generation (Simonovsky & Komodakis, 2018) and scene graph prediction (Krishna et al., 2017; Xu et al., 2017). As we have covered in this paper, most existing set predictors are either not multiset-equivariant (Vinyals et al., 2016; Stewart & Andriluka, 2016; Vaswani et al., 2017; Achlioptas et al., 2018; Li et al., 2018; Rezatofighi et al., 2020) or set-equivariant (Lee et al., 2019; Locatello et al., 2020; Kosiorek et al., 2020). The only set prediction model we are aware of that can be exclusively multiset-equivariant is DSPN (Zhang et al., 2019; Huang et al., 2020). Note that Zhang et al. (2021a) also make use of the exclusively multiset-equivariant Jacobian of sorting, but instead focus on learning multimodal densities over sets, which is tangential to our work. We provide further background in Appendix C.

One topic we have not covered in detail so far is the use of noise to separate similar or identical elements. For example, TSPN (Kosiorek et al., 2020) copies the same latent vector $z$ over all elements and adds Gaussian noise to still be able to separate them with set-equivariant self-attention. While this makes the elements different, the other problems we point out in Section 2.2 still apply, such as the fact that elements can become similar in later parts of the network and that there can be modeling problems with similar (not necessarily equal) elements. Indeed, in Section 4.1 the transformer with random position embeddings (similar application of noise as in TSPN) does not perform well.

## 6    CONCLUSION AND FUTURE WORK

We have established a more appropriate property for set prediction (exclusive multiset-equivariance) and that DSPN satisfies this property. We showed that after improving DSPN with approximate implicit differentiation, iDSPN can outperform a state-of-the-art set predictor like Slot Attention significantly. However, there are still some open problems.

**Lack of inductive bias to spatial inputs.** Most structured data can be encoded into a vector, which makes iDSPN with its vector input versatile. However, it also limits the inductive bias towards spatial domains. When working on CLEVR with image inputs and set outputs, we had to compress all spatial information into a single vector before decoding it into a set. The benefit of methods like Slot Attention is that they take another set as input, possibly including positional encoding. This allows them to work with feature map inputs rather than only feature vectors, which we believe is especially helpful with tasks like object discovery where having a spatial bias is crucial.

In preliminary experiments on the Tetrominoes object discovery task from Greff et al. (2019), we only inconsistently reached above 98% on the ARI evaluation metric, unable to match the 5-run average of 99.5% ARI for Slot Attention (Locatello et al., 2020) and 99.2% ARI for IODINE (Greff et al., 2019). We believe that in order for iDPSN to scale to larger and more complex tasks, an architecture that makes better use of spatial information is crucial. We also propose that better set prediction is a requirement for better object discovery; if a model cannot predict a set with direct supervision well, why should we expect it to be able to infer a similar one without direct supervision?

**Multiset prediction is fundamentally discontinuous.** Lastly, we made the point in Section 2 that multiset-equivariant functions like push_apart can require discontinuous changes when elements are equal. This is a concrete example of the responsibility problem (Zhang et al., 2020; Huang et al., 2020). We discuss the notion of multiset-continuity in Appendix B as a first step towards better understanding of these discontinuities. However, plenty of questions remain, for example: How can we obtain universal approximation of functions that are exclusively multiset-equivariant and multiset-continuous? How does multiset-continuity without continuity affect the training dynamics?

## REPRODUCIBILITY STATEMENT

Proofs and derivations are contained in Appendix A and Appendix D. Detailed descriptions of experiments, including how to generate these datasets in the case for Section 4.1 and Section 4.2, are given in Appendix F. We provide all the code, scripts to reproduce the experiments with the exact hyperparameters, and scripts to calculate the evaluation results at `https://github.com/davzha/multiset-equivariance`. In the code repository, we make available every training run and the corresponding pre-trained weights used to produce the results in this paper through public Weights & Biases tables (Biewald, 2020).

## ETHICS STATEMENT

We propose improvements to a general vector-to-set model and evaluate it largely on synthetic datasets. We have not yet shown the applicability of iDSPN on realistic large-scale datasets. The most immediate potential application is in object detection, which can be used for improving safety (better understanding of the world in self-driving cars) but also for surveillance (face detection and tracking). This is shared with all the general set prediction models. We showed that the main improvement on CLEVR property prediction for iDSPN is the better localization of the objects in the 3d world coordinates. This should be more useful for self-driving cars to locate objects and predict their trajectories more accurately, rather than surveillance where presence is likely more important than the precise position in 3d space.

In general, it is important to be mindful of biases when using such machine learning systems (Torralba & Efros, 2011; Mehrabi et al., 2021). For example, models trained on general object recognition datasets can be discriminatory towards lower-income households (DeVries et al., 2019). It is of course especially important to be careful when the system concerns humans, for example with biases relating to human ethnicity and gender in facial recognition systems (Klare et al., 2012; Robinson et al., 2020).

## ACKNOWLEDGMENTS

We thank the reviewers for their reviews that let us improve several aspects of the paper. We especially thank reviewer 4sjy for pointing out an issue with a definition that led us to think about it deeper and create Appendix B. The work of DWZ is part of the research programme Perspectief EDL with project number P16-25 project 3, which is financed by the Dutch Research Council (NWO) domain Applied and Engineering Sciences (TTW). This research was partially supported by Calcul Québec (`www.calculquebec.ca`), Compute Canada (`www.computecanada.ca`) and the Canada CIFAR AI Chair Program. Simon Lacoste-Julien is a CIFAR Associate Fellow in the Learning Machines & Brains program.

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

## A    THE JACOBIAN OF SORTING IS EXCLUSIVELY MULTISET-EQUIVARIANT

First, let us clarify what we mean by the Jacobian of sorting. We use $\mathtt{sort} \colon \mathbb{R}^{n \times 1} \to \mathbb{R}^{n \times 1}$ to denote sorting of a list (representing a multiset) with scalar elements in ascending order. To be precise, we define $\mathtt{sort}(\boldsymbol{X}) = \boldsymbol{S_X X}$ where $\boldsymbol{S_X}$ is the permutation matrix that sorting $\boldsymbol{X}$ would apply. $\mathtt{sort}$ in this case can be implemented by any sorting algorithm. When there are multiple possible permutations that would lead to $\boldsymbol{X}$ being sorted in the correct order (i.e. when there are duplicate elements in $\boldsymbol{X}$), $\boldsymbol{S_X}$ simply refers to the single permutation chosen by the algorithm. The details of the sorting algorithm are not relevant for the further discussion – $\boldsymbol{S_X}$ can be considered an arbitrary choice within the possible permutations that would lead to a valid sorting.

With Jacobian of sorting, we mean:

$$\frac{\partial \mathtt{sort}(\boldsymbol{X})}{\partial \boldsymbol{X}} = \frac{\partial \boldsymbol{S_X X}}{\partial \boldsymbol{X}} = \boldsymbol{S_X^\top} \tag{11}$$

In the last step, we treat $\boldsymbol{S_X}$ as independent of $\boldsymbol{X}$ because $\partial \boldsymbol{S_X} / \partial \boldsymbol{X}$ is 0 almost everywhere and undefined on a zero measure set. The definition of sorting can be extended to vector elements (the setting in FSPool (Zhang et al., 2020)) by independently sorting each dimension across the multiset so that we have a different $\boldsymbol{S_X}$ for each dimension.

### A.1    PROOF: NOT SET-EQUIVARIANT

To show that the Jacobian of numerical sorting is not set-equivariant, a simple counterexample is to realize that the Jacobian does not change between $\mathtt{sort}\left(\left[\begin{smallmatrix}1 & 0 \\ 0 & 1\end{smallmatrix}\right][0,0]^\top\right)$ and $\mathtt{sort}\left(\left[\begin{smallmatrix}0 & 1 \\ 1 & 0\end{smallmatrix}\right][0,0]^\top\right)$. The input does not change regardless of how we permute $[0,0]^\top$, therefore the Jacobian of sorting does not change.

To make this point more concrete, we implement $\mathtt{push\_apart}$ from the main text using the Jacobian of sorting. We omit the transposes in the following for brevity. We begin by defining the function $g([a,b]) = \mathtt{sort}([a,b]) \cdot [-1,1]$. This multiplies the smaller value with $-1$ and the larger with $1$ (ties broken arbitrarily), then adds them together. Taking the gradient of $g$ with respect to the input set means following the permutation that the sort applied in reverse ($\boldsymbol{S_X^\top}$), so the $-1$ gets propagated to the smaller element and $1$ to the larger one. In other words, $\nabla_{[a,b]} g = [-1,1]$ if $a \leq b$ (alternatively $a < b$), else $[1,-1]$. Now we can define $\mathtt{push\_apart}([a,b]) = [a,b] + \nabla_{[a,b]} g$ to obtain exactly the same function as in the main text.

### A.2    PROOF: MULTISET-EQUIVARIANT

We now show that the Jacobian of sorting is multiset-equivariant.

**Proposition 2.** *The Jacobian of sorting $f(\boldsymbol{X}) = \partial \mathtt{sort}(\boldsymbol{X})/\partial \boldsymbol{X} = \boldsymbol{S_X^\top}$ is multiset-equivariant.*

*Proof.* To show that for all $\boldsymbol{P}_1$, there exists a $\boldsymbol{P}_2$ that satisfies the two conditions of multiset-equivariance, we set $\boldsymbol{P}_2 = \boldsymbol{S_{P_1 X}^\top} \boldsymbol{S_X}$. We use the fact that $\boldsymbol{S}$ is a permutation matrix so $\boldsymbol{S^\top S} = \boldsymbol{S S^\top} = \boldsymbol{I}$. To show that the first condition is true:

$$\boldsymbol{P}_2 f(\boldsymbol{X}) = \boldsymbol{P}_2 \boldsymbol{S_X^\top} \tag{12}$$

$$= \boldsymbol{S_{P_1 X}^\top} \boldsymbol{S_X} \boldsymbol{S_X^\top} \tag{13}$$

$$= \boldsymbol{S_{P_1 X}^\top} \tag{14}$$

$$= f(\boldsymbol{P}_1 \boldsymbol{X}) \tag{15}$$

We now use the fact that sorting is permutation-invariant, so for any $\boldsymbol{P}$, $\boldsymbol{S_X X} = \boldsymbol{S_{PX} P X}$. To show that the second condition is true:

$$\boldsymbol{P}_2 \boldsymbol{X} = \boldsymbol{S_{P_1 X}^\top} \boldsymbol{S_X} \boldsymbol{X} \tag{16}$$

$$= \boldsymbol{S_{P_1 X}^\top} \boldsymbol{S_{P_1 X}} \boldsymbol{P}_1 \boldsymbol{X} \tag{17}$$

$$= \boldsymbol{P}_1 \boldsymbol{X} \tag{18}$$

We showed the existence of a $\boldsymbol{P}_2$ that fulfills both conditions for all $\boldsymbol{P}_1$, so $f$ is multiset-equivariant. $\qquad \square$

Note that in the definition of multiset-equivariance, it is important that the existential quantifier is on $\boldsymbol{P}_2$ acting on the output of $f$ rather than $\boldsymbol{P}_1$ acting on the input. If the quantifiers were the other way around ($\forall \boldsymbol{P}_2 \exists \boldsymbol{P}_1$), then the multiset-equivariance definition would not be useful: we have seen that permuting equal input elements does not affect the Jacobian of sorting, so there must be some $\boldsymbol{P}_2$ for which no choice of $\boldsymbol{P}_1$ works.

## B  MULTISET-CONTINUITY AND MULTISET-EQUIVARIANCE

In the main text, our discussion on the problems with separating similar elements concerns functions that are continuous and set-equivariant. In this section, we will make several aspects of this statement more precise. For example, one may ask whether there are functions that cannot separate equal elements, but can separate similar elements. This is an important point to address since exactly equal elements are rare in practice, while the behavior for similar elements is much more relevant. To talk about this properly, we first need to introduce the concept of multiset-continuity (Appendix B.1), where we use the Hungarian loss as a metric on multisets (Appendix B.2). Then, we discuss the various combinations of continuity with equivariance (Appendix B.3), and specifically that multiset-continuity combined with exclusive multiset-equivariance allows us to guarantee that not just equal elements can be separated, but also similar elements.

### B.1  MULTISET-CONTINUITY

While we used the usual notion of continuity in the main text, it is more appropriate to talk about continuity on multisets (*multiset-continuity*) when working with multiset-to-multiset functions. Multiset-continuity is desirable for a model when working with multisets for the same reasons that continuity is desirable when working with scalars or lists: it lets the model generalize from seen datapoints to nearby unseen datapoints predictably. The differing notion of continuity comes from the fact that order does not matter in multisets. This has the consequence that there are functions that are discontinuous when considered from an ordered list perspective, but they are continuous when considered from an unordered multiset perspective. The list representation is an implementation detail; the property we care about when working with multisets (regardless of how they are represented) is continuity on multisets.

Take the `push_apart` example from Section 1. With $\epsilon > 0$, `push_apart`$([0, 0]) = [-1, 1]$ but `push_apart`$([\epsilon, 0]) = [1 + \epsilon, -1]$. The list representation is discontinuous (a perturbation by $\epsilon$ changes $[-1, 1]$ to $[1 + \epsilon, -1]$, an $L_1$ distance of $4 + \epsilon$) but when considering it as a multiset, it appears continuous: $[-1, 1]$ is not much different from $[-1, 1 + \epsilon]$, the $L_1$ distance is only $\epsilon$ since we can change the "order" within the orderless multisets.

So, we would like a different notion of continuity on multisets that takes the structure of multisets into account. We can do this by using the standard $\epsilon$-$\delta$ definition of continuity on a metric space, where our metric measures distances between multisets. The multiset metric we will focus on is the Hungarian loss.

**Definition 3** (Permutation-invariant metric). A metric $D\colon \mathbb{R}^{n \times d} \times \mathbb{R}^{n \times d} \to \mathbb{R}$ is called permutation-invariant iff for any $\boldsymbol{X}, \boldsymbol{Y} \in \mathbb{R}^{n \times d}$ and any permutation matrices $\boldsymbol{P}_1, \boldsymbol{P}_2$:

$$D(\boldsymbol{X}, \boldsymbol{Y}) = D(\boldsymbol{P}_1 \boldsymbol{X}, \boldsymbol{P}_2 \boldsymbol{Y}) \tag{19}$$

**Definition 4** (Multiset-continuity). Let $(\mathbb{R}^{n \times d_1}, D_1)$ and $(\mathbb{R}^{n \times d_2}, D_2)$ be two metric spaces with permutation-invariant metrics $D_1, D_2$. A function $f\colon \mathbb{R}^{n \times d_1} \to \mathbb{R}^{n \times d_2}$ is multiset-continuous at $\boldsymbol{X} \in \mathbb{R}^{n \times d_1}$ iff for any $\epsilon > 0$, there exists a $\delta > 0$ such that $D_1(\boldsymbol{X}, \boldsymbol{Y}) < \delta$ implies $D_2(f(\boldsymbol{X}), f(\boldsymbol{Y})) < \epsilon$. The function $f$ is multiset-continuous iff it is multiset-continuous for all $\boldsymbol{X} \in \mathbb{R}^{n \times d_1}$ and we call $f$ multiset-discontinuous otherwise.

While multiple choices of permutation-invariant metrics are possible, we will use the Hungarian loss in the following. With this definition in place, the Jacobian of sorting is another example of a function that is not continuous but is multiset-continuous. It is easy to see that it is not continuous: it can only take the value of discrete permutation matrices.

**Proposition 3.** *The Jacobian of sorting $f(\boldsymbol{X}) = \partial\texttt{sort}(\boldsymbol{X})/\partial\boldsymbol{X} = \boldsymbol{S}_{\boldsymbol{X}}^\top$ is multiset-continuous.*

*Proof.* All permutation matrices $\boldsymbol{S}^\top$ are equivalent up to permutation, hence for any permutation invariant metric $D(f(\boldsymbol{X}), f(\boldsymbol{Y})) = 0$ for all $\boldsymbol{X}, \boldsymbol{Y} \in \mathbb{R}^{n \times d}$. This is always less than $\epsilon > 0$. $\qquad\square$

These definitions along with multiset-equivariance give us the necessary tools to talk about the property of being able to separate similar elements, the case most relevant in practice.

### B.2 HUNGARIAN LOSS IS A VALID METRIC

For the sake of completeness, we show that the Hungarian loss combined with an elementwise metric $D_V$ is a proper metric over multisets. The most relevant example for us is the Euclidean distance over the vector space $V = \mathbb{R}^d$, i.e., the elements of the multiset are $d$-dimensional vectors.

**Proposition 4.** *The $D_V$-Hungarian loss for two multisets $\boldsymbol{X}, \boldsymbol{Y}$ and a metric $D_V : V \times V \to \mathbb{R}$ over the elements of the multiset, is a metric:*

$$\mathcal{L}(\boldsymbol{X}, \boldsymbol{Y}) = \min_{\pi \in \Pi} \sum_i D_V(\boldsymbol{x}_i, \boldsymbol{y}_{\pi(i)}) \tag{20}$$

*Proof.* We show that $\mathcal{L}$ satisfies the three axioms:

1. Identity of indiscernibles.

$$\mathcal{L}(\boldsymbol{X}, \boldsymbol{Y}) = 0 \tag{21}$$

$$\iff \quad \min_{\pi \in \Pi} \sum_i D_V(\boldsymbol{x}_i, \boldsymbol{y}_{\pi(i)}) = 0 \tag{22}$$

$$\iff \quad \exists \pi \in \Pi : \forall i : D_V(\boldsymbol{x}_i, \boldsymbol{y}_{\pi(i)}) = 0 \tag{23}$$

$$\iff \quad \exists \pi \in \Pi : \forall i : \boldsymbol{x}_i = \boldsymbol{y}_{\pi(i)} \tag{24}$$

$$\iff \quad \boldsymbol{X} =_{\text{mset}} \boldsymbol{Y} \quad (=_{\text{mset}} \text{ denotes equality up to permutation}) \tag{25}$$

2. Symmetry.

$$\mathcal{L}(\boldsymbol{X}, \boldsymbol{Y}) = \min_{\pi \in \Pi} \sum_i D_V(\boldsymbol{x}_i, \boldsymbol{y}_{\pi(i)}) \tag{26}$$

$$= \min_{\pi \in \Pi} \sum_i D_V(\boldsymbol{y}_{\pi(i)}, \boldsymbol{x}_i) \tag{27}$$

$$= \mathcal{L}(\boldsymbol{Y}, \boldsymbol{X}) \tag{28}$$

3. Triangle inequality.

Let $\pi_{xz} = \arg\min_{\pi \in \Pi} \sum_i D_V(\boldsymbol{x}_i, \boldsymbol{z}_{\pi(i)})$ and $\pi_{zy} = \arg\min_{\pi \in \Pi} \sum_i D_V(\boldsymbol{z}_i, \boldsymbol{y}_{\pi(i)})$. These denote the optimal assignments from $\boldsymbol{X}$ to $\boldsymbol{Z}$, and from $\boldsymbol{Z}$ to $\boldsymbol{Y}$ respectively.

$$\mathcal{L}(\boldsymbol{X}, \boldsymbol{Z}) + \mathcal{L}(\boldsymbol{Z}, \boldsymbol{Y}) \tag{29}$$

$$= \min_{\pi \in \Pi} \sum_i D_V(\boldsymbol{x}_i, \boldsymbol{z}_{\pi(i)}) + \min_{\pi \in \Pi} \sum_i D_V(\boldsymbol{z}_i, \boldsymbol{y}_{\pi(i)}) \tag{30}$$

$$= \sum_i D_V(\boldsymbol{x}_i, \boldsymbol{z}_{\pi_{xz}(i)}) + \sum_i D_V(\boldsymbol{z}_i, \boldsymbol{y}_{\pi_{zy}(i)}) \qquad \pi_{xz}, \pi_{zy} \text{ are the minimizers} \tag{31}$$

$$= \sum_i D_V(\boldsymbol{x}_i, \boldsymbol{z}_{\pi_{xz}(i)}) + \sum_i D_V(\boldsymbol{z}_{\pi_{xz}(i)}, \boldsymbol{y}_{\pi_{zy}(\pi_{xz}(i))}) \qquad \text{sum is commutative} \tag{32}$$

$$= \sum_i D_V(\boldsymbol{x}_i, \boldsymbol{z}_{\pi_{xz}(i)}) + D_V(\boldsymbol{z}_{\pi_{xz}(i)}, \boldsymbol{y}_{\pi_{zy}(\pi_{xz}(i))}) \tag{33}$$

$$\geq \sum_i D_V(\boldsymbol{x}_i, \boldsymbol{y}_{\pi_{zy}(\pi_{xz}(i))}) \qquad \text{triangle inequality of } D_V \tag{34}$$

$$\geq \min_{\pi \in \Pi} \sum_i D_V(\boldsymbol{x}_i, \boldsymbol{y}_{\pi(i)}) \tag{35}$$

$$= \mathcal{L}(\boldsymbol{X}, \boldsymbol{Y}) \tag{36}$$

$$\square$$

### B.3 COMBINING MULTISET-CONTINUITY WITH EQUIVARIANCE

We now have four combinations of the properties of interest:

1. multiset-continuous and set-equivariant,
2. multiset-continuous and exclusively multiset-equivariant,
3. multiset-discontinuous and set-equivariant,
4. multiset-discontinuous and exclusively multiset-equivariant.

In the following, we show why the second combination of properties (multiset-continuous and exclusively multiset-equivariant) is most desirable.

#### B.3.1 MULTISET-CONTINUOUS AND SET-EQUIVARIANT

Most existing set-equivariant functions have this combination of properties. This is because they are continuous in the usual sense by being a composition of continuous functions, which gives us multiset-continuity.

**Proposition 5.** *If a function $f\colon \mathbb{R}^{n \times d_1} \to \mathbb{R}^{n \times d_2}$ is continuous in a metric space induced by some matrix norm, then it is also multiset-continuous.*

*Proof.* All matrix norms are equivalent in finite dimensions, thus any function that is continuous in a matrix-norm-induced metric space is also continuous in *all* such metric spaces. In particular, $f$ is also continuous for the metric $D_{\text{list}}(\boldsymbol{X}, \boldsymbol{Y}) = \sum_i^n ||\boldsymbol{x}_i - \boldsymbol{y}_i||$. We can show that $D_{\text{list}}$ is an upper bound on the permutation-invariant metric $D_{\text{mset}}$.

$$D_{\text{list}}(\boldsymbol{X}, \boldsymbol{Y}) = \sum_i^n ||\boldsymbol{x}_i - \boldsymbol{y}_i|| \tag{37}$$

$$\geq \min_{\pi \in \Pi} \sum_i^n ||\boldsymbol{x}_i - \boldsymbol{y}_{\pi(i)}|| \tag{38}$$

$$= \mathcal{L}(\boldsymbol{X}, \boldsymbol{Y}) \tag{39}$$

Since $D_{\text{list}}$ is an upper bound, if $D_{\text{list}}(f(\boldsymbol{X}), f(\boldsymbol{Y})) < \epsilon$ everywhere ($\epsilon$-$\delta$ condition of continuity), then $\mathcal{L}(f(\boldsymbol{X}), f(\boldsymbol{Y})) < \epsilon$ everywhere as well. $\qquad \square$

We know that set-equivariant models cannot separate equal elements, and by multiset-continuity, we know that a small perturbation to equal elements will only result in a small change in the multiset output. Therefore, it will be difficult to separate similar elements.

#### B.3.2 MULTISET-CONTINUOUS AND EXCLUSIVELY MULTISET-EQUIVARIANT

We have already shown that the Jacobian of sorting satisfies both multiset-continuity and exclusive multiset-equivariance. This combination of properties is really the key for separating similar elements, as we will now show.

**Proposition 6.** *Let $f\colon \mathbb{R}^{n \times d_1} \to \mathbb{R}^{n \times d_2}$ denote a multiset-to-multiset function. Exclusive multiset-equivariance is sufficient for there to exist a multiset containing equal elements that $f$ can separate.*

*Proof.* By definition, an exclusively multiset-equivariant function means that it is multiset-equivariant and not set-equivariant. Multiset-equivariance means that for all $\boldsymbol{X}$ and permutation matrices $\boldsymbol{P}_1$, there exists a permutation matrix $\boldsymbol{P}_2$ such that:

$$\boldsymbol{P}_1 \boldsymbol{X} = \boldsymbol{P}_2 \boldsymbol{X} \tag{40}$$
$$f(\boldsymbol{P}_1 \boldsymbol{X}) = \boldsymbol{P}_2 f(\boldsymbol{X}) \tag{41}$$

No set-equivariance means that there exists an $\boldsymbol{X}^\dagger$ such that $f(\boldsymbol{P}_3 \boldsymbol{X}^\dagger) \neq \boldsymbol{P}_3 f(\boldsymbol{X}^\dagger)$ for all permutation matrices $\boldsymbol{P}_3$. Combining this with Equation 41, we know that for $\boldsymbol{X}^\dagger$, we have $\boldsymbol{P}_1 \neq \boldsymbol{P}_2$. This implies that $\boldsymbol{X}^\dagger$ must contain equals to satisfy Equation 40 and that these equals become separated since $f(\boldsymbol{P}_1 \boldsymbol{X}^\dagger) = \boldsymbol{P}_2 f(\boldsymbol{X}^\dagger) \neq \boldsymbol{P}_1 f(\boldsymbol{X}^\dagger)$. Therefore, $f$ separates the equals in $\boldsymbol{X}^\dagger$. $\qquad \square$

For all exclusively multiset-equivariant functions, there exists a multiset with equal elements that can be separated. Combined with multiset-continuity, we know that similar elements around those equal elements can also be separated because a small perturbation to the equal elements (obtaining similar elements) only results in a small change in the separated output elements, so they remain separated. This highlights the appeal of multiset-continuous exclusive multiset-equivariant functions: all functions with this property can separate both equals and similar elements.

Note that while exclusive multiset-equivariance only requires there to exist a set containing equal elements that can be separated, functions like the Jacobian of sorting can actually separate *all* equals, and therefore separate all similar elements as well. This follows from the fact that each row in the permutation matrix $S^\top$ is always unique and a constant vector distance away from the other rows, regardless of how similar the input elements become.

### B.3.3  Multiset-discontinuous and set-equivariant

The common theme among the combinations with multiset-discontinuity is that we lose the guarantees we have on the behaviour for nearby points. For example, it is indeed possible to construct a function that cannot separate equals, but can separate arbitrarily similar elements. Take a `sort` function and augment it so that for equal elements, instead of resolving the tie in the permutation arbitrarily, we equally distribute the equal elements. For example, `sort2`$([1, 1]) = \left[\begin{smallmatrix} 0.5 & 0.5 \\ 0.5 & 0.5 \end{smallmatrix}\right][1, 1] = [1, 1]$. The matrix we apply to the input is no longer a permutation matrix, but a doubly-stochastic matrix. Essentially, this behaves like a mean for equal elements and like a normal sorting otherwise. The Jacobian of this operation is set-equivariant since the mean prevents equals from being separated. For arbitrarily similar elements it behaves exactly like sorting, so it can separate similar elements but it is multiset-discontinuous.

While there exist such functions that behave like a continuous exclusively multiset-equivariant function in practice (since exact equals are unlikely to be encountered), we cannot obtain any guarantees on separating similar elements in general. The following function is a counterexample to the combination of multiset-discontinuity and set-equivariance being sufficient for separating similar elements. Consider the identity function $f(X) = X$ that is augmented so that if all the entries are unique integers, it returns $X + 1$ instead. For example, $f([1, 2]) = [2, 3]$ but $f([1, 1]) = [1, 1]$ and $f([1, 2 + \epsilon]) = [1, 2 + \epsilon]$. This is multiset-discontinuous because slightly perturbing $[1, 2]$ as input makes a large change to the output in a way that cannot be compensated by the matching, and set-equivariant because equal elements are mapped to equal outputs. Applying $f$ on similar elements always just returns identity, so they can never be separated.

### B.3.4  Multiset-discontinuous and exclusively multiset-equivariant

Consider the identity function $f(X) = X$ that is augmented to separate duplicates in its input by concatenating a unique count to each duplicate element. For example, $f([1, 1, 2, 2 + \epsilon]) = [(1, 0), (1, 1), (2, 0), (2 + \epsilon, 0)]$. This function is able to separate any equals in its input through the unique count, but it is multiset-discontinuous because perturbing $[2, 2]$ to $[2, 2 + \epsilon]$ results in a large change in the output. In particular, it cannot separate arbitrarily similar elements that are not equal. This shows that exclusive multiset-equivariance alone is not enough for separating similar elements. Multiset-continuity is needed to guarantee that separation of a multiset with equal elements implies separation of that multiset with similar elements as well.

## C  Additional related work

The difference between operating on sets versus multisets has also been studied in the context of graph neighborhood aggregation (Xu et al., 2018). There, the concern is about set and multiset-*invariance* rather than equivariance: mean and max pooling lose information about duplicate elements while sum pooling does not. Equal elements easily arise in graphs (e.g. in molecules, the multiset of neighbors often contains multiple atoms of the same chemical element), so making a clear distinction between sets and multisets for invariance and equivariance properties is especially important in such contexts.

Recent works in deep learning have successfully applied implicit differentiation to large-scale language modelling (Bai et al., 2019), semantic segmentation (Bai et al., 2020), meta-learning (Rajeswaran et al., 2019), neural architecture search (Zhang et al., 2021b), and more (Agrawal et al.,

2019). The implicit function theorem allows for seamless integration of optimization problems into larger end-to-end trainable neural networks (Gould et al., 2016; Amos & Kolter, 2017; Blondel et al., 2021; El Ghaoui et al., 2021). Beyond the sets and multisets considered in this work, implicit differentiation has been applied to other non-Euclidean domains, such as hyperbolic space (Lou et al., 2020) and graphs (Gu et al., 2020).

## D  DERIVATION OF IMPLICIT DSPN

The goal of differentiating Equation 7 is to obtain the gradient of the training objective with respect to $z$ and $\theta$. If we can do that, then iDSPN fits into an autodiff framework like a normal differentiable module. We start by deriving the Jacobians for $z$ and $\theta$ with implicit differentiation and present the approximation for the Hessian afterwards. Blondel et al. (2021) describe a straightforward and efficient recipe for applying implicit differentiation to optimization problems. We follow their approach for the standard implicit differentiation derivation and efficiency concerns. We highly recommend their paper for further background and possible extensions that are made possible by implicit differentiation.

To simplify notation, we combine the target vector with the encoder parameters into $\Theta = (z, \theta)$ and restate the optimality condition:

$$\nabla_Y L(Y^*, \Theta) = \mathbf{0} \tag{42}$$

This states that the gradient of the inner loss $\nabla_Y L$ when evaluated at an optimum $(Y^*, \Theta)$ should be a matrix of zeroes. This implicitly defines a minimizer function $Y^*(\Theta)$. The implicit function theorem (further background in Krantz & Parks (2012)) states that its Jacobian $\partial Y^*(\Theta)/\partial \Theta$ exists if $\nabla_Y L$ is continuously differentiable and the Hessian $\partial(\nabla_Y L(Y^*, \Theta))/\partial Y^*$ is a square invertible matrix. Using $\nabla_Y L$ rather than $L$ lets us apply the implicit function theorem: its Jacobian is a square matrix instead of a vector and the optimality condition is fulfilled even when the root of $\nabla_Y L$ only corresponds to a local minimum of $L$. In practice, we will estimate $Y^*$ with an optimizer (e.g. gradient descent) that approximately minimizes $L$.

Using the chain rule, we differentiate both sides of Equation 42 with respect to $\Theta$ and reorder to obtain:

$$\frac{\partial \nabla_Y L(Y^*, \Theta)}{\partial Y^*} \frac{\partial Y^*(\Theta)}{\partial \Theta} = -\frac{\partial \nabla_Y L(Y^*, \Theta)}{\partial \Theta} \tag{43}$$

This is a linear system of the form $HX = B$ where we can use autodiff to compute $H$ and $B$. Solving this system for $X = \partial Y^*(\Theta)/\partial \Theta$ lets us compute the gradients for $z$ and $\theta$ with $\nabla_\Theta \mathcal{L} = (\nabla_{Y^*} \mathcal{L})^\top X$, where $\mathcal{L}$ is the loss of the prediction (as opposed to $L$).

**Efficiency.**  In order to incorporate iDSPN into an autodiff framework, we only need the vector-Jacobian product (VJP) $v^T X$ for some vector $v$. Following Blondel et al. (2021), We can efficiently compute this by solving the much smaller linear system $Hu = \nabla_{Y^*} \mathcal{L}$ for $u$. This allows us to compute the desired gradient with $\nabla_\Theta \mathcal{L} = (\nabla_{Y^*} \mathcal{L})^\top X = u^\top HX = u^\top B$.

Instead of solving the linear system exactly, it is more efficient to use the conjugate gradient method (Blondel et al., 2021). Rajeswaran et al. (2019) show that a small number of conjugate gradient steps (5 in their experiment) can be enough to obtain a sufficiently good solution. Another advantage of conjugate gradient is that we only need to compute the Hessian-vector product $Hu$ instead of the full Hessian $H$. Not only does this avoid storing $H$ explicitly (decreasing memory requirements), with reverse-mode autodiff like in PyTorch (Paszke et al., 2019) and Tensorflow (Abadi et al., 2015) it also improves parallelization: to compute the full Hessian explicitly we would have to differentiate each entry in $\nabla_Y L$ individually. The same advantage applies to $B$ since we can also use a VJP to compute $u^\top B$.

Previous uses of implicit differentiation in meta-learning (Rajeswaran et al., 2019) and neural architecture search (Zhang et al., 2021b) also involve solving a linear system, but their $Y$ corresponds to the neural network parameters and thus usually has millions of entries. In contrast, in our setting we work with a much smaller $Y$, which for example only contains 190 entries in Section 4.3.

**Regularization.**  In Rajeswaran et al. (2019), the regularization of the Hessian like we described in Section 3 appears due to regularizing $L$ by adding $\frac{\lambda}{2}||Y - Y_0||^2$. In preliminary experiments, we

observed that even if $L$ is not regularized, solving for the regularized Hessian $(\boldsymbol{H}/\gamma + \boldsymbol{I})$ yielded better performance at the same number of conjugate gradient steps. Adding the identity matrix can be seen as adding 1 to the eigenvalues to improve the conditioning of $\boldsymbol{H}$.

Of course, in the main text the efficiency improvements are irrelevant because we take $\gamma \to \infty$. Due to that, $\boldsymbol{H} = \boldsymbol{I}$ so Equation 43 simplifies to Equation 9. Fung et al. (2022) provide different motivations for this approximation.

# E    EXTENDED EXPERIMENTAL RESULTS

## E.1    CLASS-SPECIFIC NUMBERING

We perform additional experiments (Table 4) with data augmentation: by randomly permuting the input order, we can increase the effective number of training samples for the not-equivariant models. Since the other models are equivariant, data augmentation has no effect on them. We include additional runs for 8 classes (instead of 4 classes as in Section 4.1), in which we double the hidden dimensions of all models to account for the larger input. We use the same number of training steps in all runs (50,000).

We show these results in Table 4. For 4 classes, adding data augmentation restores the performance for both transformer with PE and BiLSTM in the $1\times$ and $10\times$ setting. For 8 classes, adding data augmentation does not help as much. More classes make the task more challenging, because the expected number of different permutations per input sample increases with the number of classes. Note that the classes are uniformly sampled. Increasing the number of classes while keeping the number of training steps constant means that a smaller fraction of all different permutations is seen during training.

Table 4: Extended experimental results for class-specific numbering with input set size 64 for 4 and 8 classes. Data augmentation (DA) by randomly permuting the elements for the not-equivariant models closes the performance gap to the exclusively multiset-equivariant models on 4 classes, but not for the slightly more complicated case of 8 classes. Accuracy in % (mean $\pm$ standard deviation) over 6 seeds, higher is better.

| Model | Set-equivariant | Multiset-equivariant | Training samples | | |
|---|---|---|---|---|---|
| | | | $1\times$ | $10\times$ | $100\times$ |
| *4 classes* | | | | | |
| DeepSets | ✓ | ✓ | $0.0_{\pm 0}$ | $0.0_{\pm 0}$ | $0.0_{\pm 0}$ |
| Transformer without PE | ✓ | ✓ | $0.0_{\pm 0}$ | $0.0_{\pm 0}$ | $0.0_{\pm 0}$ |
| Transformer with random PE | ✗ | ✗ | $28.5_{\pm 8.1}$ | $22.6_{\pm 5.2}$ | $22.7_{\pm 7.7}$ |
| Transformer with PE | ✗ | ✗ | $0.0_{\pm 0}$ | $46.4_{\pm 26.2}$ | $94.4_{\pm 1.8}$ |
| BiLSTM | ✗ | ✗ | $0.0_{\pm 0}$ | $25.1_{\pm 20.2}$ | $97.9_{\pm 2.1}$ |
| Transformer with random PE + DA | ✗ | ✗ | $26.83_{\pm 6.3}$ | $24.42_{\pm 4}$ | $25.52_{\pm 6.7}$ |
| Transformer with PE + DA | ✗ | ✗ | $94.93_{\pm 1.9}$ | $96.35_{\pm 1.9}$ | $96.58_{\pm 1.2}$ |
| BiLSTM + DA | ✗ | ✗ | $98.12_{\pm 0.6}$ | $99.04_{\pm 0.6}$ | $98.83_{\pm 1.6}$ |
| DSPN | ✗ | ✓ | $89.4_{\pm 2.4}$ | $88.9_{\pm 4.9}$ | $90.8_{\pm 4.2}$ |
| iDSPN (ours) | ✗ | ✓ | $96.8_{\pm 0.6}$ | $98.1_{\pm 0.3}$ | $97.9_{\pm 0.6}$ |
| *8 classes* | | | | | |
| Transformer with random PE + DA | ✗ | ✗ | $6.95_{\pm 4.5}$ | $12.03_{\pm 6.4}$ | $11.23_{\pm 2.3}$ |
| Transformer with PE + DA | ✗ | ✗ | $38.37_{\pm 12.6}$ | $96.95_{\pm 0.9}$ | $96.79_{\pm 1.7}$ |
| BiLSTM + DA | ✗ | ✗ | $61.56_{\pm 15.6}$ | $49.03_{\pm 10.4}$ | $61.67_{\pm 26.7}$ |
| DSPN | ✗ | ✓ | $72.51_{\pm 1.9}$ | $73.87_{\pm 6.8}$ | $72.65_{\pm 12.3}$ |
| iDSPN (ours) | ✗ | ✓ | $90.91_{\pm 0.6}$ | $96.33_{\pm 1.3}$ | $96.78_{\pm 1.5}$ |

E.2  RANDOM SETS AUTOENCODING

First, we show the full experimental results as mentioned in Section 4.2. We had to exclude 10 runs of DSPN due to significantly worse results making their average uninformative. We did not observe any stability issues with iDSPN, so we did not have to exclude any iDSPN runs. The following excluded DSPN runs were all using 40 iterations:

- 6 runs of $n = 2, d = 32$ (out of 8 seeds),
- 1 run of $n = 2, d = 16$,
- 2 runs of $n = 4, d = 16$,
- 1 run of $n = 4, d = 32$.

Figure 2 and Figure 3 provide different views of the same underlying data in Table 5: each subplot in Figure 2 keeps $n$ fixed and varies $d$, while each subplot in Figure 3 keeps $d$ fixed and varies $n$. Since the scale of the losses varies across the different dataset configurations, we normalize each $(n, d)$ combination by dividing the loss by the loss of the `it=10 Baseline` (DSPN) run. That is why `it=10 Baseline` always has a relative loss of 1.

The runs are re-ordered from the main text to be in the order `iDSPN with momentum, iDSPN, Baseline` to make visual comparison for equal computation easier. The correspondence between color and model is still the same. To make a comparison for equal iterations, compare the baseline DSPN (orange bar) to the two bars to the left of it (in the table, the two rows above). To make a comparison for similar computation, compare the baseline DSPN (orange bar) to the two bars to the right of it (in the table, the two rows below), but only if within the same $n, d$ dataset configuration.

**The impact of multiset-equivariance.**    We perform an additional experiment on this dataset in order to compare between an exclusively multiset-equivariant model (iDSPN with FSPool encoder and using momentum) and set-equivariant models (iDSPN with sum or mean pooling encoder and using momentum). The models and training settings are exactly the same aside from the difference in pooling method used and thus their equivariance property.

We show the results in Table 6. When comparing these results to the numbers in Table 5, it becomes clear that the exclusively multiset-equivariant iDSPN is far better. For most dataset configurations, the loss of iDSPN with sum or mean pooling are worse by a factor of 10 or more. For example, for (n=8, d=8) at 40 iterations, the loss improves from 6.4e-2 to 2.1e-3 when going from Sum to FSPool, a lower loss by a factor of around 30. Only when the task is extremely easy or extremely hard is the difference smaller: for n=2, d=2 at 20 iterations, the loss FSPool is "only" better by a factor of 1.4 (Sum 2.7e-4, FSPool 1.9e-4), and for n=32, d=32 where every model is essentially a random guess, they are almost equal (Mean 3.4e-1, FSPool 3.2e-1).

Note that we experienced significant instabilities and training divergence when using sum pooling for larger multiset sizes. We excluded these divergent runs manually (111 runs in total across all seeds), which sometimes leaves zero successful seeds for a dataset configuration (e.g. n=32, d=16 for sum pooling). We did not observe any instabilities for FSPool and mean pooling.

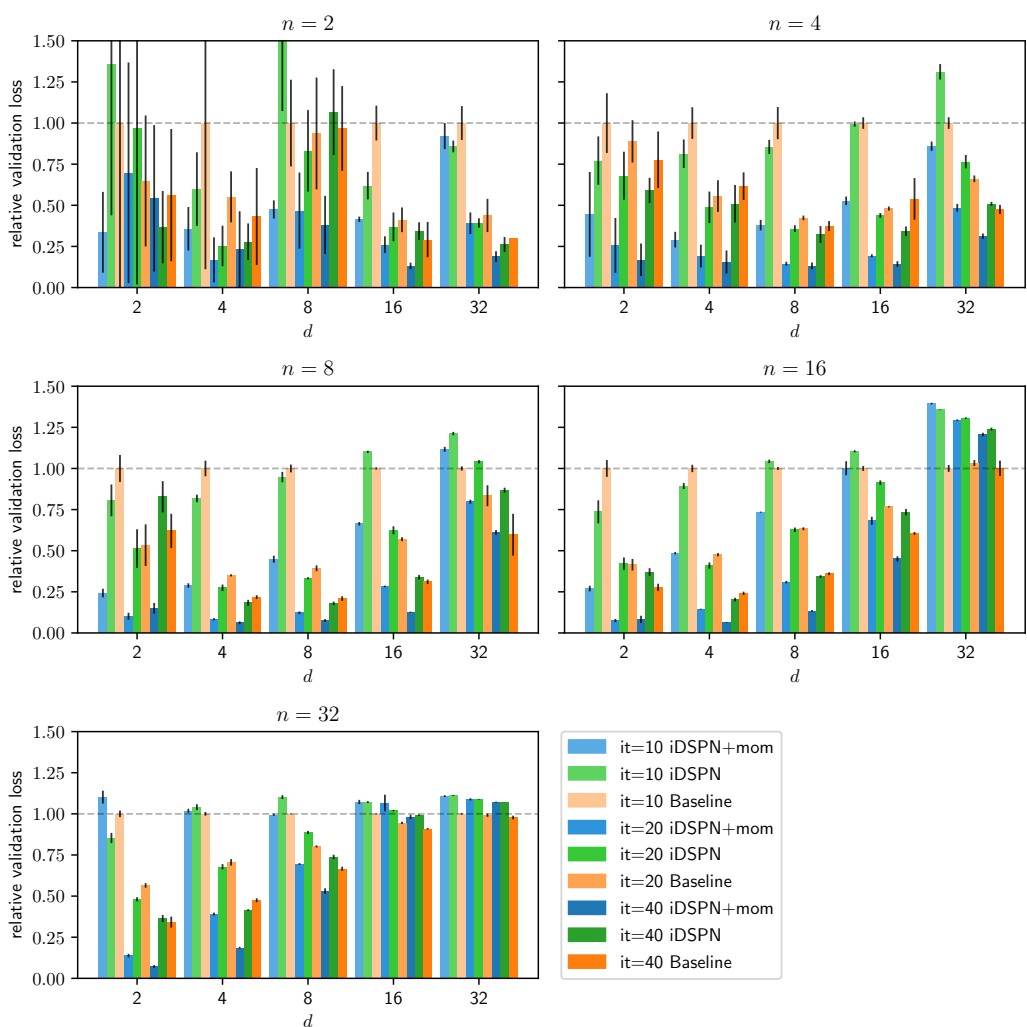

Figure 2: Relative reconstruction loss for autoencoding sets when fixing $n$ and varying $d$, loss relative to `it=10 DSPN`, lower is better.

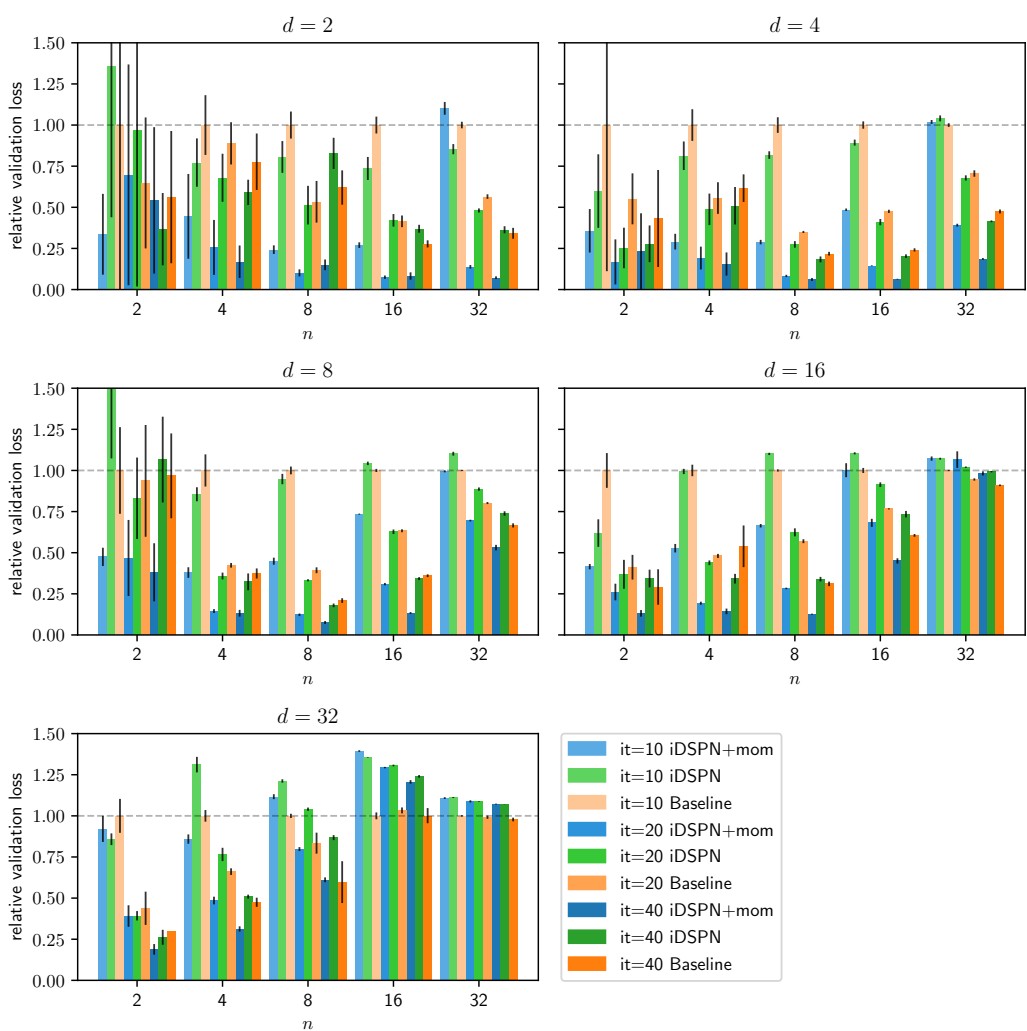

Figure 3: Relative reconstruction loss for autoencoding sets when fixing $d$ and varying $n$, loss relative to `it=10 DSPN`, lower is better.

Table 5: Full results for autoencoding sets on every dataset configuration with set size $n$ and set element dimensionality $d$, every model, and varying number of iterations. Reconstruction Huber loss (mean $\pm$ standard deviation) over 8 runs for most results, lower is better. Some dataset configurations with DSPN use fewer runs, see Appendix E.2 for the explanation.

| Model | Iterations | $d=2$ | $d=4$ | $d=8$ | $d=16$ | $d=32$ |
|---|---|---|---|---|---|---|
| | | $n=2$ | | | | |
| iDSPN+mom | 10 | $6.0\text{e-}5_{\pm4\text{e-}5}$ | $1.8\text{e-}4_{\pm6\text{e-}5}$ | $2.3\text{e-}4_{\pm3\text{e-}5}$ | $1.0\text{e-}3_{\pm4\text{e-}5}$ | $9.6\text{e-}3_{\pm8\text{e-}4}$ |
| iDSPN | 10 | $2.4\text{e-}4_{\pm2\text{e-}4}$ | $2.9\text{e-}4_{\pm1\text{e-}4}$ | $7.8\text{e-}4_{\pm2\text{e-}4}$ | $1.5\text{e-}3_{\pm2\text{e-}4}$ | $9.0\text{e-}3_{\pm4\text{e-}4}$ |
| Baseline | 10 | $1.8\text{e-}4_{\pm2\text{e-}4}$ | $4.9\text{e-}4_{\pm4\text{e-}4}$ | $4.9\text{e-}4_{\pm1\text{e-}4}$ | $2.5\text{e-}3_{\pm3\text{e-}4}$ | $1.0\text{e-}2_{\pm1\text{e-}3}$ |
| iDSPN+mom | 20 | $1.2\text{e-}4_{\pm1\text{e-}4}$ | $8.3\text{e-}5_{\pm7\text{e-}5}$ | $2.3\text{e-}4_{\pm1\text{e-}4}$ | $6.5\text{e-}4_{\pm1\text{e-}4}$ | $4.1\text{e-}3_{\pm7\text{e-}4}$ |
| iDSPN | 20 | $1.7\text{e-}4_{\pm2\text{e-}4}$ | $1.2\text{e-}4_{\pm6\text{e-}5}$ | $4.1\text{e-}4_{\pm1\text{e-}4}$ | $9.1\text{e-}4_{\pm2\text{e-}4}$ | $4.1\text{e-}3_{\pm3\text{e-}4}$ |
| Baseline | 20 | $1.2\text{e-}4_{\pm7\text{e-}5}$ | $2.7\text{e-}4_{\pm8\text{e-}5}$ | $4.6\text{e-}4_{\pm2\text{e-}4}$ | $1.0\text{e-}3_{\pm2\text{e-}4}$ | $4.6\text{e-}3_{\pm1\text{e-}3}$ |
| iDSPN+mom | 40 | $9.6\text{e-}5_{\pm8\text{e-}5}$ | $1.1\text{e-}4_{\pm1\text{e-}4}$ | $1.9\text{e-}4_{\pm9\text{e-}5}$ | $3.3\text{e-}4_{\pm5\text{e-}5}$ | $2.0\text{e-}3_{\pm3\text{e-}4}$ |
| iDSPN | 40 | $6.5\text{e-}5_{\pm4\text{e-}5}$ | $1.4\text{e-}4_{\pm5\text{e-}5}$ | $5.3\text{e-}4_{\pm1\text{e-}4}$ | $8.5\text{e-}4_{\pm1\text{e-}4}$ | $2.7\text{e-}3_{\pm5\text{e-}4}$ |
| Baseline | 40 | $1.0\text{e-}4_{\pm7\text{e-}5}$ | $2.1\text{e-}4_{\pm1\text{e-}4}$ | $4.8\text{e-}4_{\pm1\text{e-}4}$ | $7.2\text{e-}4_{\pm3\text{e-}4}$ | $3.2\text{e-}3_{\pm\text{nan}}$ |
| | | $n=4$ | | | | |
| iDSPN+mom | 10 | $3.3\text{e-}4_{\pm2\text{e-}4}$ | $5.7\text{e-}4_{\pm9\text{e-}5}$ | $2.2\text{e-}3_{\pm2\text{e-}4}$ | $9.9\text{e-}3_{\pm5\text{e-}4}$ | $5.8\text{e-}2_{\pm2\text{e-}3}$ |
| iDSPN | 10 | $5.7\text{e-}4_{\pm1\text{e-}4}$ | $1.6\text{e-}3_{\pm2\text{e-}4}$ | $4.9\text{e-}3_{\pm2\text{e-}4}$ | $1.9\text{e-}2_{\pm3\text{e-}4}$ | $8.8\text{e-}2_{\pm3\text{e-}3}$ |
| Baseline | 10 | $7.4\text{e-}4_{\pm1\text{e-}4}$ | $2.0\text{e-}3_{\pm2\text{e-}4}$ | $5.8\text{e-}3_{\pm6\text{e-}4}$ | $1.9\text{e-}2_{\pm7\text{e-}4}$ | $6.7\text{e-}2_{\pm2\text{e-}3}$ |
| iDSPN+mom | 20 | $1.9\text{e-}4_{\pm1\text{e-}4}$ | $3.8\text{e-}4_{\pm1\text{e-}4}$ | $8.4\text{e-}4_{\pm7\text{e-}5}$ | $3.6\text{e-}3_{\pm2\text{e-}4}$ | $3.3\text{e-}2_{\pm2\text{e-}3}$ |
| iDSPN | 20 | $5.1\text{e-}4_{\pm1\text{e-}4}$ | $9.6\text{e-}4_{\pm2\text{e-}4}$ | $2.1\text{e-}3_{\pm1\text{e-}4}$ | $8.3\text{e-}3_{\pm3\text{e-}4}$ | $5.2\text{e-}2_{\pm3\text{e-}3}$ |
| Baseline | 20 | $6.6\text{e-}4_{\pm1\text{e-}4}$ | $1.1\text{e-}3_{\pm2\text{e-}4}$ | $2.4\text{e-}3_{\pm8\text{e-}5}$ | $9.0\text{e-}3_{\pm2\text{e-}4}$ | $4.5\text{e-}2_{\pm1\text{e-}3}$ |
| iDSPN+mom | 40 | $1.3\text{e-}4_{\pm7\text{e-}5}$ | $3.1\text{e-}4_{\pm1\text{e-}4}$ | $7.6\text{e-}4_{\pm1\text{e-}4}$ | $2.7\text{e-}3_{\pm3\text{e-}4}$ | $2.1\text{e-}2_{\pm1\text{e-}3}$ |
| iDSPN | 40 | $4.4\text{e-}4_{\pm6\text{e-}5}$ | $1.0\text{e-}3_{\pm2\text{e-}4}$ | $1.9\text{e-}3_{\pm3\text{e-}4}$ | $6.4\text{e-}3_{\pm6\text{e-}4}$ | $3.4\text{e-}2_{\pm8\text{e-}4}$ |
| Baseline | 40 | $5.8\text{e-}4_{\pm1\text{e-}4}$ | $1.2\text{e-}3_{\pm2\text{e-}4}$ | $2.2\text{e-}3_{\pm2\text{e-}4}$ | $1.0\text{e-}2_{\pm2\text{e-}3}$ | $3.2\text{e-}2_{\pm2\text{e-}3}$ |
| | | $n=8$ | | | | |
| iDSPN+mom | 10 | $4.2\text{e-}4_{\pm5\text{e-}5}$ | $2.5\text{e-}3_{\pm1\text{e-}4}$ | $1.3\text{e-}2_{\pm6\text{e-}4}$ | $5.0\text{e-}2_{\pm8\text{e-}4}$ | $2.1\text{e-}1_{\pm3\text{e-}3}$ |
| iDSPN | 10 | $1.4\text{e-}3_{\pm2\text{e-}4}$ | $7.0\text{e-}3_{\pm2\text{e-}4}$ | $2.7\text{e-}2_{\pm9\text{e-}4}$ | $8.2\text{e-}2_{\pm5\text{e-}4}$ | $2.2\text{e-}1_{\pm2\text{e-}3}$ |
| Baseline | 10 | $1.7\text{e-}3_{\pm1\text{e-}4}$ | $8.5\text{e-}3_{\pm4\text{e-}4}$ | $2.8\text{e-}2_{\pm7\text{e-}4}$ | $7.5\text{e-}2_{\pm5\text{e-}4}$ | $1.9\text{e-}1_{\pm2\text{e-}3}$ |
| iDSPN+mom | 20 | $1.8\text{e-}4_{\pm4\text{e-}5}$ | $7.1\text{e-}4_{\pm6\text{e-}5}$ | $3.4\text{e-}3_{\pm2\text{e-}4}$ | $2.1\text{e-}2_{\pm4\text{e-}4}$ | $1.5\text{e-}1_{\pm2\text{e-}3}$ |
| iDSPN | 20 | $8.9\text{e-}4_{\pm2\text{e-}4}$ | $2.3\text{e-}3_{\pm2\text{e-}4}$ | $9.3\text{e-}3_{\pm2\text{e-}4}$ | $4.7\text{e-}2_{\pm2\text{e-}3}$ | $1.9\text{e-}1_{\pm2\text{e-}3}$ |
| Baseline | 20 | $9.3\text{e-}4_{\pm2\text{e-}4}$ | $3.0\text{e-}3_{\pm5\text{e-}5}$ | $1.1\text{e-}2_{\pm5\text{e-}4}$ | $4.3\text{e-}2_{\pm9\text{e-}4}$ | $1.5\text{e-}1_{\pm1\text{e-}2}$ |
| iDSPN+mom | 40 | $2.6\text{e-}4_{\pm6\text{e-}5}$ | $5.2\text{e-}4_{\pm8\text{e-}5}$ | $2.1\text{e-}3_{\pm2\text{e-}4}$ | $9.4\text{e-}3_{\pm2\text{e-}4}$ | $1.1\text{e-}1_{\pm3\text{e-}3}$ |
| iDSPN | 40 | $1.4\text{e-}3_{\pm2\text{e-}4}$ | $1.6\text{e-}3_{\pm2\text{e-}4}$ | $5.0\text{e-}3_{\pm3\text{e-}4}$ | $2.5\text{e-}2_{\pm1\text{e-}3}$ | $1.6\text{e-}1_{\pm3\text{e-}3}$ |
| Baseline | 40 | $1.1\text{e-}3_{\pm2\text{e-}4}$ | $1.9\text{e-}3_{\pm9\text{e-}5}$ | $5.8\text{e-}3_{\pm4\text{e-}4}$ | $2.3\text{e-}2_{\pm1\text{e-}3}$ | $1.1\text{e-}1_{\pm2\text{e-}2}$ |
| | | $n=16$ | | | | |
| iDSPN+mom | 10 | $1.0\text{e-}3_{\pm7\text{e-}5}$ | $1.1\text{e-}2_{\pm2\text{e-}4}$ | $4.4\text{e-}2_{\pm2\text{e-}4}$ | $1.5\text{e-}1_{\pm7\text{e-}3}$ | $3.0\text{e-}1_{\pm1\text{e-}3}$ |
| iDSPN | 10 | $2.8\text{e-}3_{\pm3\text{e-}4}$ | $2.0\text{e-}2_{\pm4\text{e-}4}$ | $6.2\text{e-}2_{\pm6\text{e-}4}$ | $1.7\text{e-}1_{\pm1\text{e-}3}$ | $2.9\text{e-}1_{\pm4\text{e-}4}$ |
| Baseline | 10 | $3.9\text{e-}3_{\pm2\text{e-}4}$ | $2.2\text{e-}2_{\pm5\text{e-}4}$ | $6.0\text{e-}2_{\pm6\text{e-}4}$ | $1.5\text{e-}1_{\pm2\text{e-}3}$ | $2.2\text{e-}1_{\pm4\text{e-}3}$ |
| iDSPN+mom | 20 | $2.9\text{e-}4_{\pm4\text{e-}5}$ | $3.2\text{e-}3_{\pm6\text{e-}5}$ | $1.8\text{e-}2_{\pm4\text{e-}4}$ | $1.0\text{e-}1_{\pm4\text{e-}3}$ | $2.8\text{e-}1_{\pm1\text{e-}3}$ |
| iDSPN | 20 | $1.6\text{e-}3_{\pm1\text{e-}4}$ | $9.0\text{e-}3_{\pm4\text{e-}4}$ | $3.7\text{e-}2_{\pm8\text{e-}4}$ | $1.4\text{e-}1_{\pm2\text{e-}3}$ | $2.8\text{e-}1_{\pm1\text{e-}3}$ |
| Baseline | 20 | $1.6\text{e-}3_{\pm1\text{e-}4}$ | $1.0\text{e-}2_{\pm2\text{e-}4}$ | $3.8\text{e-}2_{\pm5\text{e-}4}$ | $1.2\text{e-}1_{\pm5\text{e-}4}$ | $2.2\text{e-}1_{\pm4\text{e-}3}$ |
| iDSPN+mom | 40 | $3.2\text{e-}4_{\pm9\text{e-}5}$ | $1.4\text{e-}3_{\pm5\text{e-}5}$ | $8.0\text{e-}3_{\pm3\text{e-}4}$ | $6.9\text{e-}2_{\pm2\text{e-}3}$ | $2.6\text{e-}1_{\pm2\text{e-}3}$ |
| iDSPN | 40 | $1.4\text{e-}3_{\pm9\text{e-}5}$ | $4.5\text{e-}3_{\pm2\text{e-}4}$ | $2.0\text{e-}2_{\pm5\text{e-}4}$ | $1.1\text{e-}1_{\pm3\text{e-}3}$ | $2.7\text{e-}1_{\pm2\text{e-}3}$ |
| Baseline | 40 | $1.1\text{e-}3_{\pm8\text{e-}5}$ | $5.3\text{e-}3_{\pm2\text{e-}4}$ | $2.2\text{e-}2_{\pm5\text{e-}4}$ | $9.3\text{e-}2_{\pm1\text{e-}3}$ | $2.2\text{e-}1_{\pm1\text{e-}2}$ |
| | | $n=32$ | | | | |
| iDSPN+mom | 10 | $6.1\text{e-}3_{\pm2\text{e-}4}$ | $3.0\text{e-}2_{\pm3\text{e-}4}$ | $9.3\text{e-}2_{\pm6\text{e-}4}$ | $2.3\text{e-}1_{\pm3\text{e-}3}$ | $3.3\text{e-}1_{\pm2\text{e-}3}$ |
| iDSPN | 10 | $4.8\text{e-}3_{\pm2\text{e-}4}$ | $3.0\text{e-}2_{\pm5\text{e-}4}$ | $1.0\text{e-}1_{\pm1\text{e-}3}$ | $2.3\text{e-}1_{\pm1\text{e-}3}$ | $3.3\text{e-}1_{\pm1\text{e-}3}$ |
| Baseline | 10 | $5.6\text{e-}3_{\pm1\text{e-}4}$ | $2.9\text{e-}2_{\pm3\text{e-}4}$ | $9.3\text{e-}2_{\pm3\text{e-}4}$ | $2.2\text{e-}1_{\pm8\text{e-}4}$ | $3.0\text{e-}1_{\pm1\text{e-}3}$ |
| iDSPN+mom | 20 | $7.7\text{e-}4_{\pm6\text{e-}5}$ | $1.1\text{e-}2_{\pm2\text{e-}4}$ | $6.5\text{e-}2_{\pm5\text{e-}4}$ | $2.3\text{e-}1_{\pm1\text{e-}2}$ | $3.2\text{e-}1_{\pm2\text{e-}3}$ |
| iDSPN | 20 | $2.7\text{e-}3_{\pm7\text{e-}5}$ | $2.0\text{e-}2_{\pm5\text{e-}4}$ | $8.3\text{e-}2_{\pm9\text{e-}4}$ | $2.2\text{e-}1_{\pm1\text{e-}3}$ | $3.2\text{e-}1_{\pm6\text{e-}4}$ |
| Baseline | 20 | $3.2\text{e-}3_{\pm8\text{e-}5}$ | $2.1\text{e-}2_{\pm6\text{e-}4}$ | $7.5\text{e-}2_{\pm6\text{e-}4}$ | $2.0\text{e-}1_{\pm1\text{e-}3}$ | $3.0\text{e-}1_{\pm3\text{e-}3}$ |
| iDSPN+mom | 40 | $4.0\text{e-}4_{\pm5\text{e-}5}$ | $5.4\text{e-}3_{\pm2\text{e-}4}$ | $4.9\text{e-}2_{\pm2\text{e-}3}$ | $2.1\text{e-}1_{\pm3\text{e-}3}$ | $3.2\text{e-}1_{\pm8\text{e-}4}$ |
| iDSPN | 40 | $2.0\text{e-}3_{\pm1\text{e-}4}$ | $1.2\text{e-}2_{\pm2\text{e-}4}$ | $6.9\text{e-}2_{\pm1\text{e-}3}$ | $2.1\text{e-}1_{\pm9\text{e-}4}$ | $3.2\text{e-}1_{\pm2\text{e-}4}$ |
| Baseline | 40 | $1.9\text{e-}3_{\pm2\text{e-}4}$ | $1.4\text{e-}2_{\pm3\text{e-}4}$ | $6.2\text{e-}2_{\pm1\text{e-}3}$ | $2.0\text{e-}1_{\pm1\text{e-}3}$ | $2.9\text{e-}1_{\pm3\text{e-}3}$ |

Table 6: Autoencoding random sets on every dataset configuration with set size $n$ and set element dimensionality $d$ and varying number of iterations for set-equivariant iDSPN variants. Reconstruction Huber loss (mean ± standard deviation) over 8 runs for most results, lower is better. Some dataset configurations with iDSPN+mom+Sum use fewer runs due to divergent training behavior.

| Model | Iterations | $d = 2$ | $d = 4$ | $d = 8$ | $d = 16$ | $d = 32$ |
|---|---|---|---|---|---|---|
| | | $n = 2$ | | | | |
| iDSPN+mom+Sum | 10 | 8.3e-4±1e-4 | 1.7e-3±8e-5 | 3.5e-3±2e-4 | 9.5e-3±5e-4 | 6.4e-2±1e-2 |
| iDSPN+mom+Mean | 10 | 1.2e-3±1e-4 | 2.1e-3±2e-4 | 3.4e-3±1e-4 | 8.3e-3±2e-4 | 6.1e-2±8e-3 |
| iDSPN+mom+Sum | 20 | 2.7e-4±3e-5 | 5.7e-4±9e-5 | 1.5e-3±1e-4 | 4.8e-3±2e-4 | 2.4e-2±1e-3 |
| iDSPN+mom+Mean | 20 | 3.7e-4±1e-4 | 7.0e-4±8e-5 | 1.6e-3±9e-5 | 3.7e-3±3e-4 | 2.4e-2±1e-3 |
| iDSPN+mom+Sum | 40 | 2.9e-4±9e-5 | 5.7e-4±2e-4 | 7.5e-4±1e-4 | 3.5e-3±3e-4 | 2.1e-2±6e-4 |
| iDSPN+mom+Mean | 40 | 3.4e-4±5e-5 | 5.9e-4±3e-4 | 8.9e-4±8e-5 | 3.5e-3±3e-4 | 2.2e-2±8e-4 |
| | | $n = 4$ | | | | |
| iDSPN+mom+Sum | 10 | 4.3e-3±1e-4 | 9.9e-3±2e-4 | 3.2e-2±8e-4 | 1.1e-1±1e-3 | 2.4e-1±2e-3 |
| iDSPN+mom+Mean | 10 | 5.4e-3±4e-4 | 1.2e-2±4e-4 | 3.6e-2±5e-4 | 1.2e-1±2e-3 | 2.3e-1±1e-3 |
| iDSPN+mom+Sum | 20 | 2.7e-3±7e-4 | 3.8e-3±2e-4 | 1.6e-2±7e-4 | 1.0e-1±2e-2 | 1.9e-1±2e-3 |
| iDSPN+mom+Mean | 20 | 3.7e-3±3e-4 | 5.4e-3±3e-4 | 1.6e-2±6e-4 | 9.5e-2±3e-3 | 2.0e-1±5e-4 |
| iDSPN+mom+Sum | 40 | 3.2e-3±5e-3 | 2.3e-3±1e-4 | 1.2e-2±4e-4 | 5.5e-2±1e-2 | 1.8e-1±2e-3 |
| iDSPN+mom+Mean | 40 | 2.6e-3±3e-4 | 3.1e-3±2e-4 | 1.2e-2±3e-4 | 8.2e-2±2e-3 | 2.0e-1±8e-4 |
| | | $n = 8$ | | | | |
| iDSPN+mom+Sum | 10 | 2.5e-2±2e-2 | 5.7e-2±8e-2 | 9.5e-2±9e-4 | 2.4e-1±3e-2 | 3.1e-1±2e-3 |
| iDSPN+mom+Mean | 10 | 1.5e-2±6e-4 | 4.4e-2±4e-4 | 1.2e-1±9e-4 | 2.4e-1±4e-3 | 3.1e-1±8e-4 |
| iDSPN+mom+Sum | 20 | 4.6e-3±nan | 1.3e-2±3e-4 | 7.0e-2±9e-4 | 2.3e-1±1e-2 | 2.9e-1±9e-4 |
| iDSPN+mom+Mean | 20 | 9.9e-3±4e-4 | 2.4e-2±6e-4 | 9.6e-2±9e-4 | 2.4e-1±3e-4 | 2.9e-1±2e-3 |
| iDSPN+mom+Sum | 40 | 2.9e-3±8e-4 | 8.3e-3±3e-4 | 6.4e-2±3e-3 | 2.2e-1±5e-3 | 2.8e-1±8e-4 |
| iDSPN+mom+Mean | 40 | 6.7e-3±4e-4 | 1.6e-2±6e-4 | 8.6e-2±1e-3 | 2.4e-1±5e-4 | 2.8e-1±6e-4 |
| | | $n = 16$ | | | | |
| iDSPN+mom+Sum | 10 | 7.8e-2±3e-2 | 1.5e-1±3e-2 | 1.5e-1±2e-2 | 2.9e-1±2e-2 | 3.5e-1±2e-2 |
| iDSPN+mom+Mean | 10 | 4.2e-2±2e-3 | 1.2e-1±2e-3 | 2.1e-1±6e-3 | 2.9e-1±4e-3 | 3.5e-1±3e-3 |
| iDSPN+mom+Sum | 20 | 8.4e-3±3e-4 | 4.0e-2±5e-4 | 2.3e-1±5e-2 | 3.1e-1±3e-2 | 3.6e-1±3e-2 |
| iDSPN+mom+Mean | 20 | 2.1e-2±7e-4 | 7.1e-2±8e-4 | 1.9e-1±2e-3 | 2.8e-1±4e-4 | 3.5e-1±8e-3 |
| iDSPN+mom+Sum | 40 | 8.8e-3±2e-3 | 1.3e-1±4e-3 | 2.4e-1±1e-3 | | |
| iDSPN+mom+Mean | 40 | 1.8e-2±9e-4 | 6.5e-2±5e-4 | 1.8e-1±2e-3 | 2.7e-1±3e-4 | 3.2e-1±3e-4 |
| | | $n = 32$ | | | | |
| iDSPN+mom+Sum | 10 | 4.4e-2±2e-2 | 1.3e-1±3e-2 | 2.3e-1±1e-2 | 3.0e-1±5e-3 | 3.5e-1±6e-3 |
| iDSPN+mom+Mean | 10 | 6.3e-2±4e-3 | 1.6e-1±2e-2 | 2.6e-1±1e-2 | 2.9e-1±7e-4 | 3.5e-1±5e-4 |
| iDSPN+mom+Sum | 20 | 1.9e-2±nan | 6.8e-2±6e-4 | 1.9e-1±6e-2 | | |
| iDSPN+mom+Mean | 20 | 3.5e-2±2e-3 | 1.4e-1±1e-2 | 2.5e-1±1e-2 | 2.9e-1±1e-3 | 3.4e-1±4e-4 |
| iDSPN+mom+Sum | 40 | 6.8e-2±7e-4 | 1.6e-1±8e-4 | 2.7e-1±3e-4 | 3.4e-1±nan | |
| iDSPN+mom+Mean | 40 | 3.0e-2±4e-4 | 9.3e-2±8e-4 | 2.1e-1±9e-4 | 2.9e-1±5e-4 | 3.4e-1±2e-4 |

Table 7: Performance on CLEVR object property set prediction task, average precision (AP) in % (mean $\pm$ standard deviation) over 8 random seeds, higher is better. The first block compares iDSPN models that use different pooling methods (resulting in different equivariance properties), the second block compares iDSPN to DSPN performance in an otherwise like-to-like training setup. iDSPN + FSPool + 0.9 momentum is the same setting as shown in the main paper.

| Model | Pooling | Momentum | $AP_\infty$ | $AP_1$ | $AP_{0.5}$ | $AP_{0.25}$ | $AP_{0.125}$ | $AP_{0.0625}$ |
|---|---|---|---|---|---|---|---|---|
| iDSPN | Mean | 0.9 | $92.0_{\pm1.3}$ | $88.5_{\pm1.5}$ | $79.9_{\pm1.2}$ | $55.9_{\pm3.0}$ | $22.2_{\pm2.5}$ | $5.2_{\pm1.0}$ |
| iDSPN | Sum | 0.9 | $97.4_{\pm1.1}$ | $96.1_{\pm1.3}$ | $94.2_{\pm1.6}$ | $85.8_{\pm2.2}$ | $49.4_{\pm2.8}$ | $12.7_{\pm1.1}$ |
| iDSPN | FSPool | 0.9 | $\mathbf{98.8_{\pm0.5}}$ | $\mathbf{98.5_{\pm0.6}}$ | $\mathbf{98.2_{\pm0.6}}$ | $\mathbf{95.8_{\pm0.7}}$ | $\mathbf{76.9_{\pm2.5}}$ | $\mathbf{32.3_{\pm3.9}}$ |
| DSPN | FSPool | 0.9 | $66.6_{\pm6.0}$ | $51.9_{\pm9.7}$ | $29.9_{\pm10.8}$ | $7.8_{\pm5.0}$ | $1.4_{\pm1.0}$ | $0.0_{\pm0.0}$ |
| iDSPN | FSPool | 0.5 | $98.4_{\pm0.4}$ | $98.0_{\pm0.5}$ | $97.8_{\pm0.6}$ | $93.8_{\pm1.1}$ | $61.3_{\pm3.7}$ | $17.3_{\pm2.6}$ |
| DSPN | FSPool | 0.5 | $98.6_{\pm0.8}$ | $98.0_{\pm0.7}$ | $97.1_{\pm1.1}$ | $85.3_{\pm2.3}$ | $34.8_{\pm4.2}$ | $6.1_{\pm1.2}$ |

### E.3  CLEVR OBJECT PROPERTY PREDICTION

On CLEVR, we perform two additional experiments with 128x128 image inputs. The results for both are shown in Table 7.

**The impact of multiset-equivariance.**  First, we evaluate the difference between the exclusively multiset-equivariant iDSPN with FSPool and the set-equivariant iDSPNs with sum or mean pooling. In this setup, the models and training configurations are exactly the same except for the pooling method. The results show that the exclusively multiset-equivariant iDSPN still consistently outperforms the set-equivariant iDSPNs, though to a lesser degree than in Appendix E.2. The largest differences can still be seen for the stricter AP thresholds. This can be explained by the one-hot-encoded attributes helping in keeping set elements farther apart so that separation of similar elements is less of a concern.

**DSPN in a like-to-like setting.**  While there are various experimental differences of our iDSPN to the original DSPN setup in Zhang et al. (2019) (more details in Appendix F), we can also run DSPN in a setting as close to iDSPN as possible. This allows us to evaluate the quality of the gradient estimation by the approximate implicit differentiation compared to differentiation through the unrolled optimization. This difference in how the gradient is calculated is the only difference between the models. All other details are exactly the same as in our normal iDSPN training setup.

First, we trained DSPN with the same momentum as iDSPN, which led to significantly degraded results. This suggests that iDSPN is more stable to the inner optimization hyperparameters than DSPN. After reducing the momentum of DSPN and iDSPN from 0.9 to 0.5, we obtain more reasonable results for DSPN. Still, the iDSPN results at 0.5 momentum are *better* than the DSPN results at 0.5 momentum. Thus, we believe that the approximate implicit differentiation should not be simply viewed as trying to estimate the automatic differentiation gradient and being at best equally as good as the autodiff gradient. iDSPN outperforming DSPN in a like-to-like setting suggests that the approximate implicit gradient is actually better than the automatic differentiation gradient. This observation is inline with theoretical findings on strongly convex inner objectives (Blondel et al., 2021; Ablin et al., 2020), which indicate that implicit differentiation is better at approximating the Jacobian when the optimizer solution is inexact. Thus, iDSPN should be considered a different procedure to DSPN altogether.

Keep in mind that the better results of iDSPN in a like-to-like setting compared to DSPN also come with the other benefits of implicit differentiation such as faster training speed: in our training setup iDSPN takes around 2 hours 40 minutes to train on a V100 GPU, while DSPN takes around 3 hours 15 minutes.

# F    DETAILED EXPERIMENTAL DESCRIPTION

Unless otherwise noted, in each of the experiments, we make use of FSPool (Zhang et al., 2020) in the set encoder $g$ for iDSPN and DSPN, which allows them to be exclusively multiset-equivariant.

**Multiset loss.**    When the task is to predict sets, we need to compute a loss between a predicted and a ground-truth set. Throughout the experiments, we make use of the Hungarian loss (Kuhn, 1955):

$$\mathcal{L}(\boldsymbol{Y}^{(1)}, \boldsymbol{Y}^{(2)}) = \min_{\pi \in \Pi} \sum_i ||\boldsymbol{y}_i^{(1)} - \boldsymbol{y}_{\pi(i)}^{(2)}||^2 \tag{44}$$

It finds the optimal 1-to-1 mapping from each element in the prediction $\boldsymbol{Y}^{(1)} = [\boldsymbol{y}_1^{(1)}, \ldots, \boldsymbol{y}_n^{(1)}]^\top$ to each element in the ground-truth $\boldsymbol{Y}^{(2)} = [\boldsymbol{y}_1^{(2)}, \ldots, \boldsymbol{y}_n^{(2)}]^\top$, minimizing the total pairwise losses. A pairwise loss is a standard loss like mean squared error and cross-entropy. The Hungarian loss is permutation-invariant in both its arguments and is able to map duplicate predicted elements to different ground-truth elements and vice versa, which makes it well-suited for multiset prediction.

**Set refiner networks.**    Set refiner networks (SRN) (Huang et al., 2020) make a few minor changes to DSPN. When we mention DSPN, we use the following modifications:

- SRN found that a learning rate of 1 for the inner optimization (Equation 7) is sufficient. DSPN originally used a learning rate of 800 due to normalizing by the set size in the set encoder and a bug that made the effective learning rate depend on the batch size.
- SRN found that with the better learning rate, the additional supervised loss term that DSPN proposes is not necessary for successful training.

## F.1    CLASS-SPECIFIC NUMBERING

The following description corresponds to Section 4.1.

**Dataset.**    This experiment can be thought of as an elementwise classification that requires global information: each input element needs to be classified (predicting the numerical ID of the element), but the prediction needs to take the other predictions into account to avoid duplicate IDs.

We use a Hungarian loss to match the predictions and ground-truth within each class, because we do not care which instance of a class gets which ID as long as all the correct IDs are predicted. To match the setup in the other experiments, we can implement this as a single call to the matching algorithm by increasing the cost of pairing up two elements of different classes. We use MSE as pairwise loss for DSPN/iDSPN and cross-entropy as pairwise loss for the other models. The baselines perform worse with MSE as pairwise loss.

For each example, the input multiset has a size of 64 with 4-dimensional elements corresponding to 4 classes. This is represented as a $64 \times 4$ matrix where each row is a one-hot vector that is sampled i.i.d. from the multinomial distribution over the equally-weighted 4 classes. We generate the target multiset of the same set size with 64-dimensional elements ($64 \times 64$ matrix), each element being a one-hot vector that represents a number. For each class, we number the corresponding elements sequentially. The setting with $1\times$ samples has a training dataset of size 640. We additionally use a validation set of size 6,400 and a test set of size 64,000 for every run. In the extended experimental results in Appendix E.1 we also extend this setup to 8 classes, for which we use the same input multiset size and dataset sizes.

**Models.**    We have the prior information that every instance of a class in the input will also appear in the desired output. To make the task easier for the models, we modify them to take this into account. This makes sure that a model does not need to learn to preserve the input class in the output; it only needs to predict the ID associated to the class.

Both the equivariant DeepSets version (as opposed to the more commonly-used invariant version) and BiLSTM naturally pair up each input with each output element, so they already satisfy the desired property. If the "first" element in the input is class $c$, the "first" predicted ID in the output will naturally correspond to class $c$ as well.

For the Transformer baselines, we use an additional modification: the input set to the transformer decoder is based on an affine transformation $(\boldsymbol{Y}_0)_i = \boldsymbol{W}\boldsymbol{x}_i + \boldsymbol{b}$ of the $1 \le i \le n$ input elements. Through this, each input element is paired up with an output element as well. The Transformer without PE becomes set-equivariant to the input this way. For the Transformers with PE and random PE, we provide the position encoding only to the transformer decoder because giving it to the transformer encoder significantly degraded the performance.

We adapt the iDSPN and DSPN models to this setting by fixing the first four dimensions of $\boldsymbol{Y}$ to the values in the input multiset as we discussed in Section 3, which makes them multiset-equivariant to the input. With the set encoder $g$ architecture of Linear($4 + 64 \to 256$)–ReLU–Linear($256 \to 256$)–FSPool, they become exclusively multiset-equivariant. Since the initial set $\boldsymbol{Y}_0$ is sampled i.i.d., we do not need to pay special attention to which input element gets associated with which random initialization. During the inner optimization of iDSPN, we only update the 64 dimensions corresponding to the output set and keep the first four dimensions fixed. Additionally, we apply projected gradient descent with a `proj` that projects each element back onto a point in the probability simplex – the convex hull relaxation of the one-hot encoded IDs – after each gradient descent update. This ensures that the vectors in $\boldsymbol{Y}$ remain valid probability distributions after each update. We found that this significantly improved the performance over the unconstrained version.

**Evaluation.** We compute the accuracy at the sample-level, meaning a predicted set is considered correct only if every element is correct. The predicted ID for each predicted element is obtained by taking the argmax over the elements' dimensions in the output. The baselines are very volatile during training, which results in very large variances at the end of training. To reduce this variance, we pick the best model according to the validation accuracy, evaluated every 500 training steps. We report the accuracy on the test set for the best model. Each run samples new datasets based on the random seed, so we evaluate all models using the same set of random seeds.

## F.2 RANDOM SETS AUTOENCODING

The following description corresponds to Section 4.2.

**Dataset.** Every set has a fixed size with $n$ elements of dimensionality $d$. Every element is sampled i.i.d. from $\mathcal{N}(\boldsymbol{0}, \boldsymbol{I})$. We assign 64,000 samples to the training set and 6,400 samples to the test set. Changing the random seed also changes the dataset. As loss and evaluation metric, we use Hungarian matching with the Huber loss (Huber, 1964) (quadratic for distances below 1, linear for distances above 1) as pairwise loss. We always use the results for a model after the final epoch without any early-stopping (we did not observe overfitting) or selection based on best loss.

**Models.** We try to make the DSPN baseline and the iDSPN models as similar as possible. Since DSPN was intended to be used with a learned initial set $\boldsymbol{Y}_0$ (Zhang et al., 2019), we match this in iDSPN and add the regularizer $0.1||\boldsymbol{Y} - \boldsymbol{Y}_0||^2$ to Equation 8 as we discussed in Section 3. We apply the same regularization to the objective in DSPN to make the objectives match up. We initialize $\boldsymbol{Y}_0$ at the start of training with elements sampled from $\mathcal{N}(\boldsymbol{0}, \boldsymbol{I}/10)$ for all models. For the same seed, the weight initializations are the same for all models.

We use the same set encoder to encode the autoencoder input between the models. We use the same set encoder $g$ between the models (but we do not share the weights with the encoder for the autoencoder input). The architecture of these set encoders is Linear($d \to 512$)–ReLU–Linear($512 \to 512$)–FSPool, where the linear layers are applied on each element in a multiset individually with shared weights across the elements.

Note that we do not compare against Slot Attention here due to the difference in setup. Slot Attention takes a set directly as input, so giving it the input set directly would not give us a permutation-invariant bottleneck, which somewhat defeats the point of the autoencoder. It only has to learn to produce the identity function, which we verified it can easily do.

**Training.** All models are trained for 40 epochs at batch size 128 with Adam (Kingma & Ba, 2015) with a learning rate of 1e-3 and default momentum parameters. This is enough to converge for almost every model and dataset combination. Only the hardest difficulty settings ($32 \times 16, 16 \times 32, 32 \times 32$) would show any benefit from more training.

### F.3 CLEVR OBJECT PROPERTY PREDICTION

The following description corresponds to Section 4.3. Where possible, we use exactly the same setup as DSPN.

Every image is resized to 128x128 (or 256x256) and scaled to be in the interval $[0, 1]$. For the ground-truth, we scale down the target 3d coordinates from the interval $[-3, 3]$ to $[-1, 1]$ and concatenate it to the attributes (each of which is one-hot encoded) and the binary mask feature (1 if element is present, 0 if not). The input image is processed by a ResNet encoder, which scales the input down to a 4x4x512 (8x8x512) feature map, on which we apply BatchNorm followed by a 2x2 convolution with stride 2 to obtain a 2x2x128 (4x4x32) feature map. This is flattened to a latent vector with 512 dimensions, which is the input $z$ to iDSPN. We use the Hungarian loss with Huber loss (Huber, 1964) as pairwise loss. We train the model for 100 epochs with the Adam optimizer (Kingma & Ba, 2015), a learning rate of 1e-3, and default momentum hyperparameters.

For evaluation, we adapt the average precision evaluation code to PyTorch. It is calculated by sorting the predicted elements in descending order by the dimension corresponding to the binary mask feature (acting as confidence), then computing the precision-recall curve by including more and more predicted elements, starting from the most confident elements. A prediction is considered correct only if all attributes are correct (after taking the argmax for each attribute individually) and the 3d coordinate is within a given threshold.

We make some changes to the training procedure of DSPN:

- Instead of using a relation network (Santoro et al., 2017) – expanding the set into the set of all pairs first – paired with FSPool, we skip the relation network entirely. This improves the complexity of the set encoder $g$ used in iDSPN from $O(n^2 \log n)$ to $O(n \log n)$. Using the relation network approach would improve our results slightly (e.g. 3.5 percentage points improvement on $AP_{0.125}$ for $128 \times 128$), but is also a bit slower. For simplicity, we therefore opted to not using relation networks.

  The architecture of the set encoder $g$ in iDSPN is Linear($19 \rightarrow 512$)–ReLU–Linear($512 \rightarrow 512$)–FSPool. The main difference to DSPN is that since there is no concatenation of pairs, so the input dimensionality is 19 instead of 38 and everything is applied on sets of size $n$ rather than sets of size $n^2$.

- Instead of ResNet34 to encode the input image, we use the smaller ResNet18. This did not appear to affect results.

- Instead of using a learned initial set $Y_0$ as in DSPN, we find that it makes no difference to randomly sample the initial set for every example. We therefore use the latter for simplicity and to match the Slot Attention setup. In initial experiments we found that even initializing every element to $\mathbf{0}$ causes no problems.

  While we do not use the initialization with 0s in our experiments to better match the setup in Slot Attention, there are benefits to this initialization. In Slot Attention, the variance of the random initialization is learned, but the variance must strike a balance between being too low (elements are more similar, which poses difficulty in separating them) and too high (model has to perform well over a wider value range, which is more difficult). Exclusively multiset-equivariant models can avoid this trade-off since separating similar elements poses no problem. This is another benefit of exclusive multiset-equivariance.

- We increase the batch size from 32 to 128. There appeared to be no difference in results between the two, with 128 being faster by making better use of parallelization.

- We use Nesterov's Accelerated Gradient (Nesterov, 1983) with a momentum parameter of 0.9 instead of standard gradient descent without momentum.

- Instead of fixing the number of iterations at 10 like DSPN, we set the number of iterations to 20 at the start of training and change it to 40 after 50 epochs. This had slightly better training loss than starting training with 40 iterations. We have tried a few other ways of increasing the number of iterations throughout training (going from 10 to 20 to 30 to 40 iterations, smooth increase from 1 to 40 over the epochs, randomly sampling an iteration between 20 and 40 every batch), which had little impact on results. iDSPN training was stable in all of these configurations.

- We drop the learning rate after 90 epochs from 1e-3 to 1e-4 for the last 10 epochs. This slightly improved training loss while also reducing variance in epoch-to-epoch validation loss.

- In preliminary experiments, we rarely observed spikes in the training loss. Clipping the gradients in the inner optimization to a maximum L2 norm of 10 seemed to help.

Note that we use a ResNet18 image encoder, while Slot Attention uses a simpler 4-layer convolutional neural network with $5 \times 5$ kernels. This difference is not so easily compared: we compress the image into a 512d vector while they operate on a feature map. Their final feature map for CLEVR has 32x32 spatial positions with 64d each, so it can be argued that their latent space is 65536-dimensional. It is natural that a tighter bottleneck requires more processing. ResNet18 also applies several strided convolutions early, so the amount of processing between it and the Slot Attention image encoder are not too dissimilar. This is reflected by the fact that the time taken to process a sample is similar between ResNet18 + iDSPN and CNN + Slot Attention, even though we use a smaller batch size and could gain a speed-up from better parallelization for higher batch sizes.

A small difference between the default Slot Attention setup and our setup is that they normalize the 3d coordinates to be within [0, 1] while we use an interval of [-1, 1]. In Table 3, Slot Attention* uses an interval of [-1, 1] (the same as iDSPN) while Slot Attention† uses the interval of [-3, 3] (the default coordinate range for this dataset). This increases the weight on the coordinates versus the other attributes, so it improves AP for strict thresholds but trades off classification performance at a threshold of infinity. We did not experiment with different coordinate scales and simply kept it the same as DSPN.

We observe some overfitting with 128x128 images (1.6e-4 train loss, 5.4e-4 val loss), which did not appear in preliminary experiments when training an autoencoder with the ground-truth set as input. We reduce this generalization gap by increasing the image size to 256x256 while keeping the latent vector size the same, which results in further performance improvements (1.1e-4 train loss, 2.5e-4 val loss). We thus believe that the overfitting is due to the ResNet18 image encoder rather than iDSPN.

## G    EXAMPLE INPUTS AND OUTPUTS

In the following, we show some example inputs of each dataset and corresponding model outputs. The focus is not on showing representative examples, but highlighting some qualitative differences.

Table 8 shows the worst failure case of each model trained with $100\times$ samples when numbering elements within a class (Section 4.1). For each model, the sample is selected from the dataset by picking the prediction with the highest number of misclassifications. Since we make every model equivariant to the input, the classes shown in the prediction are guaranteed to be correct; the difficulty lies only in numbering them correctly. Note that the classes in the input are ordered arbitrarily and the order of the numberings within each class is arbitrary as well; we sort the outputs in the table so that it is easier to tell what elements are missing or duplicated. Interestingly, the worst input for transformer with PE, BiLSTM, DSPN, and iDSPN all contain a class that only occurs twice, which is atypically few.

Figure 4 shows the autoencoding of randomly-generated sets (not in training set nor validation set) for each $n$ and $d$ combination (Section 4.2). We leave out iDSPN without momentum because its performance is so similar to DSPN. Both models apply 20 iterations in the forward pass. The visual difference between model qualities is often somewhat subtle and best seen when zooming in on a digital display. It is most obvious on $(n = 4, d = 16)$ and $(n = 8, d = 16)$, but there are smaller differences visible with configurations like $(n = 32, d = 2)$ as well, where iDSPN with momentum tends to be slightly more centered on the ground-truth than DSPN. The examples where $n \times d \geq 512$ show that even if DSPN has a better loss than iDSPN on those configurations, the model quality is far from acceptable.

Table 9 and Table 10 show predictions on CLEVR for different number of evaluation iterations (Section 4.3). We checked the first 128 images of the CLEVR validation set manually and selected examples that looked difficult due to occluded objects, then evaluated the model on these examples. For easier examples, iDSPN is usually correct on all objects at 40 iterations. The mistakes of the model are reasonable, such as confusing a yellow cube that has its sides and bottom occluded for a cylinder (third image). The improvement seen when going from 128x128 to 256x256 images is also quite natural, even for a human observer.

Table 8: Worst failure cases for numbering elements within a class, selected by the highest amount of wrong numberings in one prediction. Predictions are sorted for better readability. Wrong numberings are marked in **red**.

| DeepSets | Transformer without PE | Transformer with random PE | Transformer with PE | BiLSTM | DSPN | iDSPN |
|---|---|---|---|---|---|---|
| (a, 0) | (a, 4) | (a, 0) | (a, 0) | (a, 0) | (a, 0) | **(a, 2)** |
| **(a, 2)** | (a, 4) | (a, 1) | (a, 1) | (a, 1) | (a, 1) | **(a, 3)** |
| **(a, 2)** | **(a, 4)** | **(a, 1)** | (a, 2) | (a, 2) | (a, 2) | (b, 0) |
| **(a, 2)** | **(a, 4)** | (a, 2) | (a, 3) | (a, 3) | (a, 3) | (b, 1) |
| **(a, 2)** | **(a, 4)** | (a, 3) | (a, 4) | **(a, 3)** | (a, 4) | (b, 2) |
| **(a, 2)** | **(a, 4)** | (a, 4) | **(a, 4)** | (a, 4) | (a, 5) | (b, 3) |
| **(a, 2)** | **(a, 4)** | (a, 6) | (a, 5) | (a, 5) | (a, 6) | (b, 4) |
| **(a, 2)** | **(a, 4)** | (a, 7) | (a, 7) | (a, 6) | (a, 7) | (b, 5) |
| **(a, 2)** | **(a, 4)** | (a, 8) | (a, 8) | (a, 7) | (a, 8) | (b, 6) |
| **(a, 2)** | **(a, 4)** | (a, 9) | (a, 9) | (a, 8) | (a, 9) | (b, 7) |
| **(a, 2)** | **(a, 4)** | (a, 10) | (a, 10) | (a, 9) | (a, 10) | (b, 8) |
| **(a, 2)** | **(a, 4)** | (a, 11) | (a, 11) | (a, 11) | (a, 11) | (b, 9) |
| **(a, 2)** | **(a, 4)** | (a, 12) | (a, 12) | (a, 12) | (a, 12) | (b, 10) |
| (b, 1) | (b, 0) | (b, 0) | (a, 13) | (a, 13) | (a, 13) | (b, 11) |
| **(b, 1)** | **(b, 0)** | **(b, 0)** | (a, 14) | **(b, 3)** | **(b, 4)** | (b, 12) |
| **(b, 1)** | **(b, 0)** | (b, 1) | (a, 15) | **(b, 3)** | **(b, 6)** | (b, 13) |
| **(b, 1)** | **(b, 0)** | (b, 2) | (a, 16) | (c, 0) | (c, 0) | (c, 0) |
| **(b, 1)** | **(b, 0)** | (b, 4) | (a, 17) | (c, 1) | (c, 1) | (c, 1) |
| **(b, 1)** | **(b, 0)** | (b, 5) | (a, 18) | (c, 2) | (c, 2) | (c, 2) |
| **(b, 1)** | **(b, 0)** | (c, 0) | (a, 19) | (c, 3) | (c, 3) | (c, 3) |
| **(b, 1)** | **(b, 0)** | (c, 1) | (a, 21) | (c, 4) | (c, 4) | (c, 4) |
| **(b, 1)** | **(b, 0)** | (c, 2) | (a, 22) | **(c, 4)** | (c, 5) | (c, 5) |
| **(b, 1)** | **(b, 0)** | (c, 3) | (a, 26) | (c, 5) | (c, 6) | (c, 6) |
| **(b, 1)** | **(b, 0)** | (c, 4) | (a, 27) | (c, 7) | **(c, 6)** | **(c, 6)** |
| **(b, 1)** | **(b, 0)** | (c, 5) | **(a, 27)** | (c, 8) | (c, 7) | (c, 7) |
| **(b, 1)** | **(b, 0)** | (c, 6) | (a, 28) | **(c, 8)** | (c, 8) | (c, 8) |
| (c, 5) | (c, 4) | (c, 7) | **(a, 28)** | (c, 9) | **(c, 8)** | (c, 9) |
| **(c, 5)** | **(c, 4)** | (c, 8) | **(a, 28)** | (c, 11) | (c, 9) | (c, 10) |
| **(c, 5)** | **(c, 4)** | (c, 9) | **(a, 28)** | (c, 12) | (c, 10) | (c, 11) |
| **(c, 5)** | **(c, 4)** | (c, 10) | (a, 29) | (c, 13) | **(c, 10)** | (c, 12) |
| **(c, 5)** | **(c, 4)** | (c, 11) | **(a, 29)** | (d, 0) | (c, 11) | (c, 13) |
| **(c, 5)** | **(c, 4)** | (c, 12) | **(a, 29)** | (d, 1) | (c, 12) | (c, 14) |
| **(c, 5)** | **(c, 4)** | (c, 13) | **(a, 29)** | (d, 2) | (c, 13) | (c, 15) |
| **(c, 5)** | **(c, 4)** | (c, 14) | **(a, 29)** | (d, 4) | (c, 14) | (c, 16) |
| **(c, 5)** | **(c, 4)** | (d, 0) | (b, 0) | (d, 5) | (c, 15) | (c, 17) |
| **(c, 5)** | **(c, 4)** | (d, 2) | (b, 1) | (d, 6) | (c, 16) | (c, 18) |
| **(c, 5)** | **(c, 4)** | (d, 3) | (b, 2) | (d, 7) | (c, 17) | (c, 19) |
| **(c, 5)** | **(c, 4)** | (d, 4) | (b, 3) | (d, 10) | **(c, 17)** | (c, 20) |
| **(c, 5)** | **(c, 4)** | (d, 5) | (b, 4) | (d, 11) | (c, 18) | (c, 22) |
| **(c, 5)** | **(c, 4)** | **(d, 5)** | (b, 5) | (d, 12) | (c, 19) | (c, 23) |
| **(c, 5)** | **(c, 4)** | **(d, 5)** | **(b, 5)** | (d, 13) | (c, 20) | (c, 24) |
| **(c, 5)** | **(c, 4)** | (d, 6) | (b, 6) | (d, 15) | **(c, 20)** | (c, 25) |
| (d, 14) | (d, 0) | (d, 7) | (b, 7) | (d, 16) | (c, 21) | (c, 26) |
| **(d, 14)** | **(d, 0)** | (d, 8) | (b, 8) | (d, 17) | (c, 22) | **(c, 26)** |
| **(d, 14)** | **(d, 0)** | (d, 9) | (b, 9) | (d, 19) | (c, 23) | **(c, 26)** |
| **(d, 14)** | **(d, 0)** | (d, 10) | (b, 10) | (d, 20) | (c, 24) | (c, 27) |
| **(d, 14)** | **(d, 0)** | (d, 11) | (b, 11) | (d, 22) | (c, 26) | (c, 28) |
| **(d, 14)** | **(d, 0)** | **(d, 11)** | (b, 12) | (d, 23) | (c, 27) | (c, 29) |
| **(d, 14)** | **(d, 0)** | (d, 14) | (c, 0) | (d, 24) | **(c, 27)** | **(c, 29)** |
| **(d, 14)** | **(d, 0)** | (d, 15) | (c, 1) | (d, 25) | (c, 28) | **(c, 29)** |
| **(d, 14)** | **(d, 0)** | (d, 17) | (c, 2) | (d, 26) | (d, 0) | (d, 0) |
| **(d, 14)** | **(d, 0)** | (d, 18) | (c, 3) | **(d, 26)** | (d, 1) | (d, 1) |
| **(d, 14)** | **(d, 0)** | **(d, 18)** | (c, 4) | (d, 27) | (d, 2) | (d, 2) |
| **(d, 14)** | **(d, 0)** | (d, 19) | (c, 5) | (d, 28) | (d, 3) | (d, 3) |
| **(d, 14)** | **(d, 0)** | (d, 20) | (c, 6) | **(d, 28)** | (d, 4) | (d, 4) |
| **(d, 14)** | **(d, 0)** | **(d, 20)** | (c, 7) | **(d, 28)** | (d, 5) | (d, 5) |
| **(d, 14)** | **(d, 0)** | (d, 24) | (c, 8) | **(d, 28)** | (d, 6) | (d, 6) |
| **(d, 14)** | **(d, 0)** | (d, 25) | (c, 9) | (d, 29) | (d, 7) | (d, 7) |
| **(d, 14)** | **(d, 0)** | **(d, 25)** | (c, 10) | **(d, 29)** | (d, 8) | (d, 8) |
| **(d, 14)** | **(d, 0)** | **(d, 25)** | (c, 11) | **(d, 29)** | (d, 9) | (d, 9) |
| **(d, 14)** | **(d, 0)** | (d, 27) | (c, 12) | **(d, 29)** | (d, 10) | (d, 10) |
| **(d, 14)** | **(d, 0)** | **(d, 27)** | **(c, 12)** | **(d, 29)** | (d, 11) | (d, 11) |
| **(d, 14)** | **(d, 0)** | **(d, 27)** | **(d, 4)** | **(d, 29)** | (d, 12) | (d, 12) |
| **(d, 14)** | **(d, 0)** | (d, 27) | **(d, 5)** | **(d, 29)** | (d, 13) | (d, 13) |

Figure 4: Random set examples for each combination of $n$ and $d$. For $d > 2$, only the first two dimensions are shown. All models use 20 iterations.

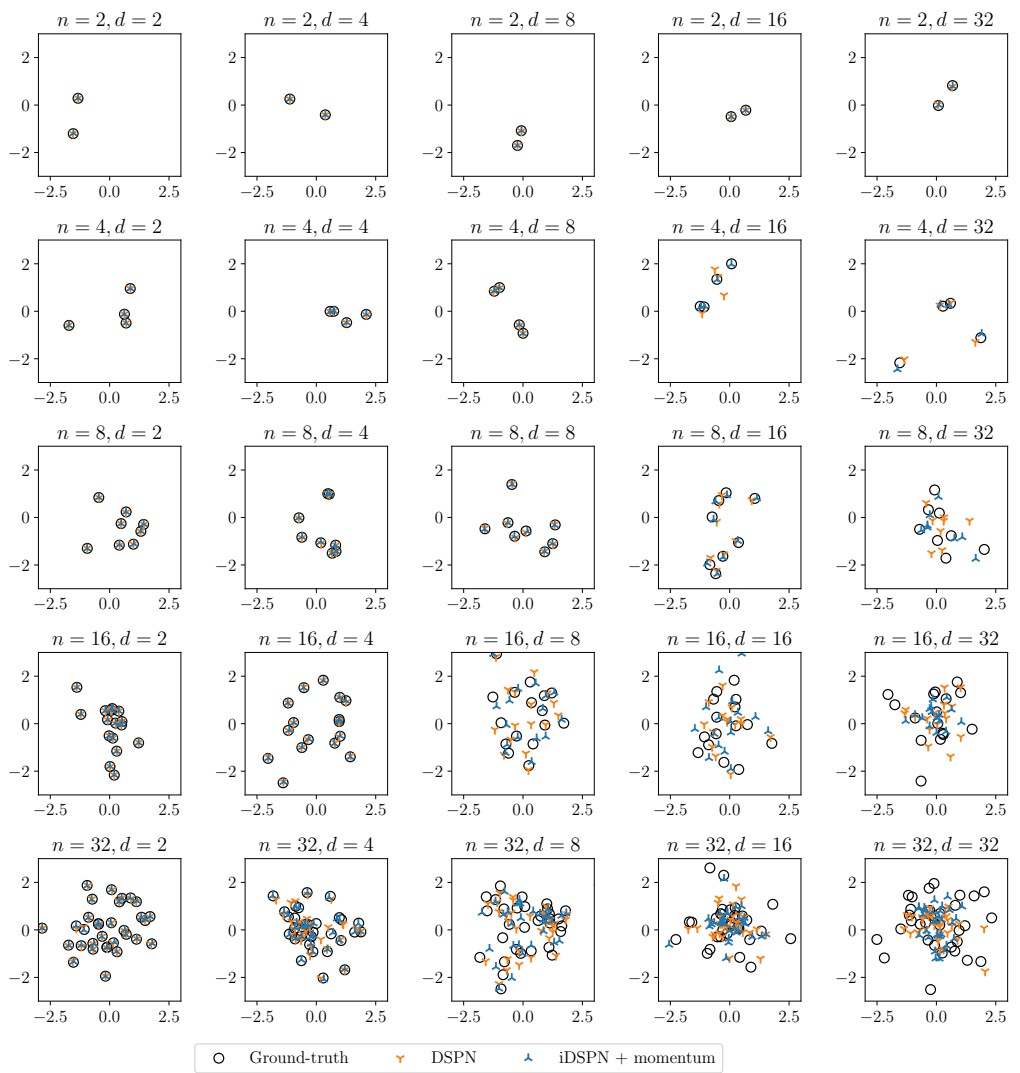

Table 9: CLEVR examples with 128x128 images, cherry-picked by difficulty (highly-occluded objects). Incorrect attributes and Euclidean distances (d) greater than 0.25 marked in **red**. Ground-truth objects are ordered going from top left to bottom right diagonally, predicted objects are matched to that ordering based on Equation 44. We choose an iDSPN run with median performance and evaluate at different iterations.

| Image | Ground-truth | iDSPN 10 iterations | iDSPN 20 iterations | iDSPN 40 iterations |
|---|---|---|---|---|
|  | (-2.95 -2.56 0.70) large purple metal cylinder
(-2.26 0.66 0.35) small cyan metal sphere
(-2.13 1.62 0.35) small brown metal cylinder
(-0.83 -1.80 0.35) small red rubber sphere
(-0.34 -2.86 0.35) small cyan rubber cylinder
(0.37 2.25 0.70) large cyan rubber cube
(1.25 -2.94 0.70) large yellow rubber cylinder
(1.41 0.07 0.35) small purple metal cylinder
(2.02 2.29 0.35) small brown metal cylinder
(2.93 -2.16 0.70) large red metal cube | (-2.79 -2.19 0.64) **d=0.41** large purple metal cylinder
(-2.48 0.60 0.26) d=0.24 small cyan metal sphere
(-1.46 2.06 -0.06) **d=0.90** small brown metal cylinder
(-0.69 -2.33 0.35) **d=0.55** small **yellow** rubber sphere
(0.45 -2.73 0.62) **d=0.84** small **yellow** rubber cylinder
(0.73 0.96 0.44) **d=1.36** **small red** rubber cube
(0.55 -2.16 -0.49) **d=1.59** large **brown metal** cylinder
(0.59 0.13 -0.12) **d=0.95** small purple metal cylinder
(2.04 2.23 -0.04) **d=0.40** small brown metal cylinder
(3.15 -2.34 0.68) **d=0.29** large red metal cube | (-2.85 -2.58 0.79) d=0.13 large purple metal cylinder
(-2.32 0.41 0.41) **d=0.26** small cyan metal sphere
(-2.04 1.70 0.41) d=0.14 small brown metal cylinder
(-0.81 -1.84 0.32) d=0.05 small red rubber sphere
(-0.08 -2.73 0.43) **d=0.30** small **yellow** rubber cylinder
(0.44 2.24 0.70) d=0.06 large cyan rubber cube
(1.04 -2.36 0.84) **d=0.63** large yellow rubber cylinder
(1.04 0.05 0.46) **d=0.39** small purple metal cylinder
(1.97 2.21 0.46) d=0.14 small brown metal cylinder
(2.96 -2.34 0.83) d=0.23 large red metal cube | (-2.89 -2.53 0.67) d=0.07 large purple metal cylinder
(-2.31 0.45 0.33) d=0.21 small cyan metal sphere
(-2.03 1.74 0.37) d=0.16 small brown metal cylinder
(-0.81 -1.83 0.33) d=0.04 small red rubber sphere
(-0.24 -2.86 0.33) d=0.11 small cyan rubber cylinder
(0.42 2.28 0.68) d=0.06 large cyan rubber cube
(1.30 -2.90 0.72) d=0.07 large yellow rubber cylinder
(1.09 0.10 0.31) **d=0.33** small purple metal cylinder
(1.96 2.26 0.38) d=0.07 small brown metal cylinder
(2.90 -2.20 0.70) d=0.05 large red metal cube |
|  | (-1.73 2.38 0.35) small green metal cylinder
(-1.21 -2.81 0.35) small red metal cylinder
(-0.77 2.09 0.35) small brown rubber sphere
(0.10 -2.88 0.70) large purple metal sphere
(0.38 -0.02 0.70) large blue rubber cube
(1.25 2.14 0.35) small cyan metal cylinder
(1.62 -2.93 0.35) small yellow rubber cube
(2.30 0.40 0.70) large gray metal sphere | (-1.52 2.75 0.31) **d=0.42** small green metal cylinder
(-1.22 -2.60 0.21) **d=0.25** small **purple** metal cylinder
(-0.14 2.12 0.36) **d=0.63** small brown rubber **cylinder**
(0.48 -2.71 0.32) **d=0.57** large purple metal sphere
(-0.08 -0.79 0.74) **d=0.90** large **gray metal sphere**
(0.62 0.55 0.31) **d=1.70** small **blue** metal cylinder
(1.46 -2.83 0.31) d=0.18 small yellow rubber cube
(2.03 0.63 0.84) **d=0.37** large gray metal sphere | (-1.58 2.64 0.47) **d=0.32** small green metal cylinder
(-1.34 -2.82 0.35) d=0.13 small red metal cylinder
(-0.15 2.05 0.34) **d=0.62** small brown rubber **cylinder**
(0.23 -2.83 0.79) d=0.16 large purple metal sphere
(0.12 0.02 0.60) **d=0.28** large blue rubber cube
(0.83 1.08 0.40) **d=1.14** small **blue** metal **cube**
(1.57 -2.90 0.39) d=0.06 small yellow rubber cube
(2.34 0.39 0.78) d=0.09 large gray metal sphere | (-1.73 2.48 0.37) d=0.10 small green metal cylinder
(-1.32 -2.82 0.30) d=0.12 small red metal cylinder
(-0.49 1.76 0.35) **d=0.44** small brown rubber **cylinder**
(0.16 -2.83 0.66) d=0.09 large purple metal sphere
(0.29 -0.05 0.71) d=0.09 large blue rubber cube
(1.20 2.15 0.32) d=0.06 small cyan metal cylinder
(1.58 -2.87 0.33) d=0.07 small yellow rubber cube
(2.26 0.39 0.69) d=0.04 large gray metal sphere |
|  | (-2.73 -0.68 0.70) large cyan rubber sphere
(-2.09 1.53 0.70) large blue rubber cube
(-2.00 -2.71 0.70) large cyan rubber sphere
(-1.69 2.84 0.35) small yellow metal cube
(-0.18 2.91 0.70) large gray rubber cube
(0.12 -2.40 0.35) small blue metal cylinder
(0.75 1.26 0.70) large yellow metal cylinder
(0.91 -0.23 0.35) small blue metal cylinder
(1.97 -2.56 0.70) large red rubber cube | (-2.60 0.45 0.75) **d=1.13** large cyan rubber sphere
(-1.03 -0.97 -0.61) **d=3.02** **small cyan metal cylinder**
(-2.27 -2.48 0.30) **d=0.54** large cyan rubber sphere
(-0.58 2.04 0.25) **d=1.36** **large** yellow metal **cylinder**
(-0.45 3.01 0.64) **d=0.29** large gray rubber cube
(0.03 -1.84 0.31) **d=0.57** small blue metal cylinder
(0.99 -0.14 0.58) **d=1.42** large **blue** metal cylinder
(0.43 0.90 -0.23) **d=1.35** small blue metal cylinder
(2.00 -2.79 0.76) d=0.24 large red rubber cube | (-2.59 -0.04 0.70) **d=0.66** large cyan rubber sphere
(-1.92 1.28 0.51) **d=0.36** large blue rubber cube
(-2.23 -2.80 0.83) **d=0.28** large cyan rubber sphere
(-1.33 2.81 0.55) **d=0.41** small yellow metal **cylinder**
(-0.34 3.02 0.73) d=0.20 large gray rubber cube
(-0.01 -2.30 0.38) d=0.17 small blue metal cylinder
(0.71 0.91 0.72) **d=0.35** large **blue** metal cylinder
(0.77 -0.29 0.43) d=0.17 small blue metal cylinder
(1.98 -2.63 0.67) d=0.08 large red rubber cube | (-2.66 -0.38 0.71) **d=0.30** large cyan rubber sphere
(-2.15 1.55 0.70) d=0.07 large blue rubber cube
(-2.14 -2.80 0.69) d=0.16 large cyan rubber sphere
(-1.47 2.75 0.36) d=0.24 small yellow metal **cylinder**
(-0.29 2.98 0.67) d=0.13 large gray rubber cube
(0.06 -2.17 0.37) d=0.24 small blue metal cylinder
(0.87 1.16 0.68) d=0.16 large yellow metal cylinder
(0.78 -0.49 0.36) **d=0.29** small blue metal cylinder
(1.95 -2.65 0.68) d=0.10 large red rubber cube |
|  | (-2.82 2.97 0.70) large cyan rubber cylinder
(-2.28 -2.62 0.70) large purple metal cylinder
(-1.71 -0.51 0.35) small cyan metal cylinder
(-1.45 2.38 0.35) small yellow rubber cylinder
(0.00 -1.35 0.35) small red rubber cylinder
(0.43 2.64 0.70) large purple metal cylinder
(1.32 -1.70 0.70) large brown rubber cylinder
(2.41 0.23 0.35) small red metal sphere | (-2.86 2.82 0.78) d=0.18 large cyan rubber cylinder
(-2.34 -2.70 0.70) d=0.10 large purple metal cylinder
(-1.02 -0.54 0.63) **d=0.74** small cyan metal cylinder
(-1.72 2.36 0.41) **d=0.27** small yellow rubber cylinder
(0.22 -0.88 0.59) **d=0.57** small **brown** rubber cylinder
(0.42 2.57 0.69) d=0.07 large purple metal cylinder
(1.23 -1.69 0.67) d=0.10 large brown rubber cylinder
(2.36 0.33 0.10) **d=0.27** small red metal sphere | (-2.84 3.01 0.71) d=0.04 large cyan rubber cylinder
(-2.26 -2.65 0.81) d=0.12 large purple metal cylinder
(-1.74 -0.63 0.34) d=0.12 small cyan metal cylinder
(-1.52 2.36 0.39) d=0.08 small yellow rubber cylinder
(0.24 -1.00 0.34) **d=0.42** small **brown** rubber cylinder
(0.45 2.62 0.70) d=0.03 large purple metal cylinder
(1.22 -1.83 0.72) d=0.16 large brown rubber cylinder
(2.44 0.22 0.39) d=0.04 small red metal sphere | (-2.85 2.98 0.70) d=0.03 large cyan rubber cylinder
(-2.28 -2.63 0.71) d=0.01 large purple metal cylinder
(-1.67 -0.69 0.34) d=0.19 small cyan metal cylinder
(-1.58 2.43 0.34) d=0.14 small yellow rubber cylinder
(0.22 -1.00 0.35) **d=0.41** small red rubber cylinder
(0.46 2.62 0.67) d=0.05 large purple metal cylinder
(1.25 -1.80 0.72) d=0.12 large brown rubber cylinder
(2.44 0.25 0.36) d=0.04 small red metal sphere |
|  | (-2.87 -2.91 0.35) small gray metal sphere
(-2.78 2.14 0.35) small purple metal cube
(-2.33 -1.03 0.70) large brown rubber cylinder
(-1.73 -2.79 0.35) small gray rubber cube
(-1.31 2.74 0.35) small blue metal cube
(-1.18 1.67 0.35) small brown metal sphere
(-0.01 2.36 0.70) large blue rubber cube
(0.45 -1.73 0.70) large gray rubber cube | (-2.81 -2.86 0.40) d=0.09 small gray metal sphere
(-2.61 2.22 0.49) d=0.23 small purple metal cube
(-2.00 -0.62 0.57) **d=0.54** large brown rubber cylinder
(-1.61 -2.20 0.72) **d=0.71** **large** gray rubber cube
(-0.48 0.65 0.41) **d=2.26** small **green rubber** cube
(-1.43 2.43 0.47) **d=0.81** small brown metal sphere
(-0.06 2.36 0.60) d=0.11 large blue rubber cube
(0.06 -2.07 0.51) **d=0.55** **small** gray rubber cube | (-2.80 -2.82 0.46) d=0.15 small gray metal sphere
(-2.70 2.33 0.29) d=0.21 small purple metal cube
(-2.36 -0.88 0.76) d=0.17 large brown rubber cylinder
(-1.73 -2.71 0.40) d=0.10 small gray rubber cube
(-1.15 1.80 0.25) **d=0.96** small **gray** metal **sphere**
(-1.13 1.89 0.38) d=0.23 small brown metal sphere
(0.01 2.47 0.66) d=0.12 large blue rubber cube
(0.50 -1.76 0.82) d=0.13 large gray rubber cube | (-2.83 -2.88 0.33) d=0.05 small gray metal sphere
(-2.72 2.35 0.42) d=0.22 small purple metal cube
(-2.42 -0.95 0.73) d=0.13 large brown rubber cylinder
(-1.64 -2.59 0.34) d=0.22 small gray rubber cube
(-1.20 2.24 0.34) **d=0.52** small **green** metal **sphere**
(-1.17 1.60 0.36) d=0.06 small brown metal sphere
(0.01 2.36 0.69) d=0.02 large blue rubber cube
(0.49 -1.76 0.68) d=0.05 large gray rubber cube |

Table 10: CLEVR examples with 256x256 images, cherry-picked by difficulty (highly-occluded objects), same examples as in Table 9. Incorrect attributes and Euclidean distances (d) greater than 0.25 marked in red. Ground-truth objects are ordered going from top left to bottom right diagonally, predicted objects are matched to that ordering based on Equation 44. We choose an iDSPN run with median performance and evaluate at different iterations.

| Image | Ground-truth | iDSPN 10 iterations | iDSPN 20 iterations | iDSPN 40 iterations |
|---|---|---|---|---|
|  | (-2.95 -2.56 0.70) large purple metal cylinder | (-2.70 -1.53 0.53) **d=1.07** **small** purple metal **sphere** | (-2.11 -2.49 0.69) **d=0.84** large **red rubber** cylinder | (-3.01 -2.50 0.69) d=0.09 large purple metal cylinder |
| | (-2.26 0.66 0.35) small cyan metal sphere | (-2.43 0.59 0.38) d=0.18 small **red** metal **cylinder** | (-2.51 -0.48 0.46) **d=1.18** small **purple** metal sphere | (-2.22 0.56 0.35) d=0.11 small cyan metal sphere |
| | (-2.13 1.62 0.35) small brown metal cylinder | (-1.11 1.51 0.35) **d=1.02** small brown metal cylinder | (-2.28 1.80 0.31) d=0.23 small brown metal cylinder | (-2.20 1.60 0.35) d=0.08 small brown metal cylinder |
| | (-0.83 -1.80 0.35) small red rubber sphere | (0.01 -2.17 0.14) **d=0.94** small **cyan** rubber **cylinder** | (-0.68 -2.02 0.38) **d=0.26** small **cyan** rubber sphere | (-0.78 -1.76 0.36) d=0.06 small red rubber sphere |
| | (-0.34 -2.86 0.35) small cyan rubber cylinder | (0.84 -2.20 0.59) **d=1.37** small **yellow** rubber cylinder | (0.02 -0.97 0.36) **d=1.93** small cyan rubber cylinder | (-0.37 -2.61 0.34) d=0.25 small cyan rubber **cube** |
| | (0.37 2.25 0.70) large cyan rubber cube | (0.26 1.52 0.69) **d=0.74** large **brown** rubber **cylinder** | (0.36 2.32 0.70) d=0.07 large cyan rubber cube | (0.40 2.30 0.69) d=0.06 large cyan rubber cube |
| | (1.25 -2.94 0.70) large yellow rubber cylinder | (-1.12 -2.51 0.75) **d=2.41** large **red** rubber cylinder | (0.94 -2.98 0.48) **d=0.38** large yellow rubber cylinder | (1.24 -3.02 0.73) d=0.09 large yellow rubber cylinder |
| | (1.41 0.07 0.35) small purple metal cylinder | (0.32 0.48 0.46) **d=1.16** small **cyan** metal **cube** | (1.44 -0.32 0.41) **d=0.40** small purple metal cylinder | (1.45 0.03 0.36) d=0.06 small purple metal cylinder |
| | (2.02 2.29 0.35) small brown metal cylinder | (2.34 2.00 0.35) **d=0.43** small brown metal cylinder | (1.97 2.27 0.36) d=0.06 small brown metal cylinder | (2.02 2.24 0.34) d=0.05 small brown metal cylinder |
| | (2.93 -2.16 0.70) large red metal cube | (2.89 -2.53 0.77) **d=0.38** large red metal cube | (2.86 -2.41 0.66) **d=0.26** large red metal cube | (2.85 -2.21 0.68) d=0.10 large red metal cube |
|  | (-1.73 2.38 0.35) small green metal cylinder | (-1.53 2.32 0.40) d=0.21 small green metal cylinder | (-1.03 1.59 0.14) **d=1.07** small green metal cylinder | (-1.79 2.43 0.37) d=0.08 small green metal cylinder |
| | (-1.21 -2.81 0.35) small red metal cylinder | (-1.18 -2.85 0.37) d=0.05 small red metal cylinder | (-1.23 -2.78 0.33) d=0.04 small red metal cylinder | (-1.18 -2.83 0.34) d=0.04 small red metal cylinder |
| | (-0.77 2.09 0.35) small brown rubber sphere | (0.19 0.38 0.54) **d=1.97** small brown **metal** sphere | (-1.10 2.05 0.32) **d=0.33** small brown rubber sphere | (-0.70 2.00 0.35) d=0.12 small brown rubber sphere |
| | (0.10 -2.88 0.70) large purple metal sphere | (0.01 -2.44 0.53) **d=0.48** large purple metal sphere | (0.13 -2.85 0.71) d=0.04 large purple metal sphere | (0.14 -2.90 0.70) d=0.04 large purple metal sphere |
| | (0.38 -0.02 0.70) large blue rubber cube | (0.28 0.16 0.60) d=0.22 large blue rubber cube | (0.36 0.25 0.64) **d=0.28** large blue rubber cube | (0.39 0.09 0.69) d=0.11 large blue rubber cube |
| | (1.25 2.14 0.35) small cyan metal cylinder | (0.81 2.32 0.51) **d=0.50** small cyan metal cylinder | (1.06 2.23 0.39) d=0.22 small cyan metal cylinder | (1.23 2.12 0.35) d=0.03 small cyan metal cylinder |
| | (1.62 -2.93 0.35) small yellow rubber cube | (1.76 -3.24 0.37) **d=0.34** small yellow rubber cube | (1.68 -2.91 0.36) d=0.07 small yellow rubber cube | (1.68 -2.96 0.35) d=0.07 small yellow rubber cube |
| | (2.30 0.40 0.70) large gray metal sphere | (2.23 0.19 0.80) d=0.24 large gray metal sphere | (2.28 0.41 0.74) d=0.05 large gray metal sphere | (2.30 0.35 0.70) d=0.05 large gray metal sphere |
|  | (-2.73 -0.68 0.70) large cyan rubber sphere | (-2.71 -1.83 0.69) **d=1.15** large cyan rubber sphere | (-2.75 -0.59 0.70) d=0.08 large cyan rubber sphere | (-2.76 -0.69 0.70) d=0.03 large cyan rubber sphere |
| | (-2.09 1.53 0.70) large blue rubber cube | (-2.55 1.25 0.66) **d=0.54** large blue rubber cube | (-2.06 1.56 0.69) d=0.04 large blue rubber cube | (-2.04 1.54 0.71) d=0.05 large blue rubber cube |
| | (-2.00 -2.71 0.70) large cyan rubber sphere | (-0.67 -1.94 0.57) **d=1.55** large cyan **metal** sphere | (-2.09 -2.71 0.68) d=0.09 large cyan rubber sphere | (-2.05 -2.69 0.69) d=0.06 large cyan rubber sphere |
| | (-1.69 2.84 0.35) small yellow metal cube | (-1.35 2.36 0.30) **d=0.59** small yellow metal **cylinder** | (-1.56 2.78 0.37) d=0.15 small yellow metal **cylinder** | (-1.58 2.80 0.38) d=0.11 small yellow metal **cylinder** |
| | (-0.18 2.91 0.70) large gray rubber cube | (-0.03 2.21 0.69) **d=0.72** large gray rubber cube | (-0.17 2.87 0.71) d=0.04 large gray rubber cube | (-0.18 2.89 0.69) d=0.03 large gray rubber cube |
| | (0.12 -2.40 0.35) small blue metal cylinder | (0.62 -1.77 0.12) **d=0.84** small blue metal cylinder | (0.26 -2.31 0.31) d=0.17 small blue metal cylinder | (0.14 -2.34 0.36) d=0.07 small blue metal cylinder |
| | (0.75 1.26 0.70) large yellow metal cylinder | (-0.29 2.44 0.57) **d=1.58** large yellow metal **cube** | (0.76 1.36 0.73) d=0.11 large yellow metal cylinder | (0.74 1.31 0.70) d=0.05 large yellow metal cylinder |
| | (0.91 -0.23 0.35) small blue metal cylinder | (0.74 0.47 0.45) **d=0.72** small blue metal cylinder | (0.88 -0.36 0.38) d=0.14 small blue metal cylinder | (1.01 -0.37 0.34) d=0.18 small blue metal cylinder |
| | (1.97 -2.56 0.70) large red rubber cube | (1.85 -2.34 0.69) **d=0.25** large red rubber cube | (1.99 -2.52 0.71) d=0.04 large red rubber cube | (1.98 -2.55 0.70) d=0.01 large red rubber cube |
|  | (-2.82 2.97 0.70) large cyan rubber cylinder | (-2.92 3.07 0.55) d=0.21 large cyan rubber cylinder | (-2.78 3.02 0.76) d=0.08 large cyan rubber cylinder | (-2.82 2.92 0.69) d=0.05 large cyan rubber cylinder |
| | (-2.28 -2.62 0.70) large purple metal cylinder | (-1.86 -1.51 0.53) **d=1.20** large purple metal cylinder | (-1.69 -2.70 0.69) **d=0.60** large purple metal cylinder | (-2.24 -2.56 0.70) d=0.08 large purple metal cylinder |
| | (-1.71 -0.51 0.35) small cyan metal cylinder | (-0.65 -0.98 0.17) **d=1.17** small **brown** metal cylinder | (-1.77 -0.41 0.52) d=0.20 small cyan metal cylinder | (-1.78 -0.47 0.37) d=0.09 small cyan metal cylinder |
| | (-1.45 2.38 0.35) small yellow rubber cylinder | (-1.31 2.00 0.47) **d=0.41** small yellow rubber cylinder | (-1.26 2.29 0.34) d=0.21 small yellow rubber cylinder | (-1.39 2.42 0.35) d=0.07 small yellow rubber cylinder |
| | (0.00 -1.35 0.35) small red rubber cylinder | (1.11 -1.28 0.49) **d=1.11** small red rubber cylinder | (-0.78 -0.85 0.31) **d=0.93** small **cyan** rubber cylinder | (0.07 -1.32 0.32) d=0.07 small red rubber cylinder |
| | (0.43 2.64 0.70) large purple metal cylinder | (0.30 2.58 0.60) d=0.17 large purple metal cylinder | (0.50 2.71 0.68) d=0.10 large purple metal cylinder | (0.42 2.73 0.69) d=0.09 large purple metal cylinder |
| | (1.32 -1.70 0.70) large brown rubber cylinder | (-1.04 -2.39 0.59) **d=2.46** large **purple metal** cylinder | (1.04 -1.48 0.51) **d=0.41** **small red** rubber cylinder | (1.30 -1.76 0.68) d=0.06 large brown rubber cylinder |
| | (2.41 0.23 0.35) small red metal sphere | (2.04 0.36 0.48) **d=0.42** small red metal sphere | (2.37 0.17 0.36) d=0.08 small red metal sphere | (2.41 0.22 0.35) d=0.01 small red metal sphere |
|  | (-2.87 -2.91 0.35) small gray metal sphere | (-2.44 -2.11 0.39) **d=0.91** small gray metal sphere | (-2.86 -2.92 0.36) d=0.01 small gray metal sphere | (-2.87 -2.93 0.35) d=0.02 small gray metal sphere |
| | (-2.78 2.14 0.35) small purple metal cube | (-2.46 2.17 0.19) **d=0.36** small purple metal cube | (-2.76 2.06 0.35) d=0.09 small purple metal cube | (-2.80 2.15 0.37) d=0.03 small purple metal cube |
| | (-2.33 -1.03 0.70) large brown rubber cylinder | (-1.72 -1.06 0.56) **d=0.63** large brown rubber cylinder | (-2.26 -0.98 0.64) d=0.11 large brown rubber cylinder | (-2.26 -0.98 0.68) d=0.10 large brown rubber cylinder |
| | (-1.73 -2.79 0.35) small gray rubber cube | (-2.30 -2.98 0.52) **d=0.62** small gray **metal** cube | (-1.78 -2.73 0.39) d=0.08 small gray rubber cube | (-1.81 -2.78 0.34) d=0.08 small gray rubber cube |
| | (-1.31 2.74 0.35) small blue metal cube | (-0.96 2.02 0.21) **d=0.81** small blue metal cube | (-1.43 2.26 0.50) **d=0.52** small blue metal cube | (-1.38 2.16 0.34) **d=0.59** small blue metal cube |
| | (-1.18 1.67 0.35) small brown metal sphere | (-1.67 1.90 0.48) **d=0.56** small brown metal sphere | (-1.12 1.72 0.35) d=0.08 small brown metal sphere | (-1.14 1.72 0.34) d=0.06 small brown metal sphere |
| | (-0.01 2.36 0.70) large blue rubber cube | (-0.46 2.39 0.67) **d=0.46** large blue rubber cube | (-0.17 2.39 0.71) d=0.16 large blue rubber cube | (-0.22 2.45 0.71) d=0.23 large blue rubber cube |
| | (0.45 -1.73 0.70) large gray rubber cube | (0.15 -1.75 0.68) **d=0.30** large gray rubber cube | (0.40 -1.72 0.72) d=0.06 large gray rubber cube | (0.44 -1.68 0.69) d=0.06 large gray rubber cube |

