# OpenReview forum: "Multiset-Equivariant Set Prediction with Approximate Implicit Differentiation"
_ICLR.cc/2022/Conference — ICLR 2022 Poster_

### Official Review · Reviewer_MvMj · 2021-10-29

**Correctness:** 3
**Technical Novelty And Significance:** 3
**Empirical Novelty And Significance:** 3
**Recommendation:** 8
**Confidence:** 4

**Main Review:**

- The paper is very well written and easy to understand. It is also correct as far as I can tell (apart from a tiny thing I noticed below).
- Technically, it builds on top of the nicely carved out, yet somewhat subtle difference of multiset-equivariance and set-equivariance. The proposed improvement of DSPN with implicit differentiation is somewhat incremental: it combines existing ideas from implicit differentiation with an existing method, i.e. DSPN.
- The experimental section illustrates large improvements in sample efficiency (e.g. vs. BiLSTMs or Transformers) on designed toy data. Also on already used data (CLEVR as in Locatello et al), significant improvements are demonstrated. As this is the only more realistic dataset where improvements are shown, it would be great to extend the analysis to another dataset.

Minor Issues.
- in eqs 8,9,10 : \nabla_Y L(Y*,...) is trivially zero; what is meant is likely either \nabla_{Y*} or L(Y,...) ?
- pg. 8 Results. "Slot Attention\dagger" is not clear what is meant from the main text. Please refer to Table 3.

**Summary Of The Paper:**

The paper "Multiset-equivariant set prediction with approximate implicit differentiation" points out a difference between typically looked at set equivariance and multiset equivariance, the latter being able to produce different outputs for identical inputs. It is further shown that a method from the literature (DSPN) is multiset equivariant; to make this method perform better, this method is improved with an implicit differentiation approach. The superiority of the method is demonstrated on toy data, and also on CLEVR.

**Summary Of The Review:**

The paper is well written, nicely connects a novel theoretical observation with a modeling improvement idea, which then culminates into a clear empirical improvement. More empirical investigations would be great, but I feel the paper is well rounded as is.

---

> ### Author Response · Authors · 2021-11-11
> **Response to reviewer MvMj**
>
> **As this is the only more realistic dataset where improvements are shown, it would be great to extend the analysis to another dataset.**
> While experiment 2 does not appear representative at first glance, many real-world tasks can be considered easier instances of it (due to global structure, as we explain in section 4.2). And please refer to our response to reviewer PV3V for some additional experiments we are performing. They are on the same datasets as currently in the paper, but they make the evidence for multiset-equivariance and iDSPN more extensive.
>
> **in eqs 8,9,10 :** $\nabla_Y L(Y^*,...)$ **is trivially zero; what is meant is likely either** $\nabla_{Y^*}$ **or** $L(Y,...)$**?**
>
> Equation 8 and 10 describe the optimality condition that the minimizer $Y^*(z, \theta)$ has to fulfill. For example, equation 8 is read as “the gradient of $L$ wrt its argument $Y$, when evaluated at $Y^*$, $z$, and $\theta$, is equal to zero. $\nabla_Y L$ is the gradient (see Eq 6, where $L$ is defined in terms of $Y$), the following $(Y^*, z, \theta)$ is the evaluation of that gradient at a point.
>
> Eq 9 denotes the gradients of the implicitly defined function $Y^*(z, \theta)$ w.r.t. $z$ and $\theta$. The $\nabla_Y L(Y^*,z,\theta)$ here can be understood as a function of $z$ and $\theta$.
> We will extend the Appendix with further descriptions on the notation used in eq 8-10.
>
> **pg. 8 Results. "Slot Attention$\dagger$" is not clear what is meant from the main text. Please refer to Table 3.**
> The difference is briefly mentioned in the caption of Table 3. We will make this difference more explicit in the main text. They are two slightly different configurations of slot attention: in the training loss, Slot Attention$\dagger$ puts even more weight on getting the 3d coordinates accurate over getting the properties correct compared to Slot Attention*. That’s why when the 3d coordinates don’t matter with AP$_\infty$, the results are slightly worse for the dagger version, but when the 3d coordinates are more important, the dagger version has better AP.

---

### Official Review · Reviewer_KAp5 · 2021-11-02

**Correctness:** 4
**Technical Novelty And Significance:** 3
**Empirical Novelty And Significance:** 4
**Recommendation:** 6
**Confidence:** 4

**Main Review:**

## Strengths

* The proposed methodology of using multiset equivariance makes sense and is a meaningful difference from standard set equivariance. This is a very good catch, as the difference is subtle but meaningful.
* The empirical results are very strong. In particular, the results in the tables 2 and 3 show that the proposed model is able to actually learn (instead of being stuck at 0)
* The contribution of using implicit differentiation for the DSPN is very useful for the field.

## Weaknesses

* The motivation for multiset equivariance is often confusing and/or very handwavy. Specifically, the push_apart function only serves to show what multiset equivariance is, but I'm not sure how this motivates any real world example. If possible, having a more real-world example would be much better. Similarly, the "the problem with set-equivariance" paragraph also suffers from a lack of specificity. A simple toy experiment here would be very helpful.

## Nitpicks

* The figures should be ordered correctly and stretch across too many pages. Currently, the text talks about, in order, the results in figure 1, table 2, and table 3. However, the actual figures appear in the order of table 2, figure 1, table 3. Furthermore, the figures do not necessarily all need to be at the top of a new page, especially as table 2 pushes too far away from its source text.
* The paper should include a related work section discussing more implicit differentiation works. In addition to classical deep implicit layers works like [1, 2], I also recommend implicit layers work on other non-Euclidean domains beyond sets such as [3, 4].
* The difference between the two "Slot Attention" rows in table 3 is not made clear.

## References
[1] https://arxiv.org/abs/1607.05447

[2] https://arxiv.org/abs/1910.12430

[3] https://arxiv.org/abs/2003.00335

[4] https://arxiv.org/abs/2009.06211

**Summary Of The Paper:**

The paper proposes the usage of multiset equivariance as a weaker constraint for deep sets. The authors then modify DSPN to be multiset equivariant and introduce the usage of Jacobian-free implicit differentiation to speed up computation. Finally, the authors compare on two test tasks and show general improvements.

**Summary Of The Review:**

Overall, I found the paper to be a strong addition to the deep sets literature with many important contributions. However, I did have some overall complaints about the general structure given in the weaknesses and nitpicks sections above. As such, I am a bit reticent to accept. If these issues are resolved, I am more than happy to increase my rating.

---

> ### Author Response · Authors · 2021-11-11
> **Response to reviewer KAp5**
>
> **The motivation for multiset equivariance is often confusing and/or very handwavy. Specifically, the push_apart function only serves to show what multiset equivariance is, but I'm not sure how this motivates any real world example. If possible, having a more real-world example would be much better.**
> Exclusive multiset-equivariance allows a function to map equal elements to different outputs and most importantly makes separating similar elements easier. This can be useful for example in point cloud autoencoding tasks (e.g. Achlioptas et al, 2018; Luo, Shitong, and Wei Hu, 2021), which experiment 2 is a harder instance of. In this real-world setting, we would want to move the 3d points in the point cloud from some initial position (often randomly sampled, so little control over how close together any pair of points is) to form some shape in the end, like a chair or table. The ability to separate similar elements is important to prevent points from clumping together midway without any way to push them apart again.
> We will mention this real-world example to motivate multiset-equivariance.
>
> [Luo, Shitong, and Wei Hu. "Diffusion probabilistic models for 3d point cloud generation." In CVPR, 2021.]
>
>
> **Similarly, the "the problem with set-equivariance" paragraph also suffers from a lack of specificity. A simple toy experiment here would be very helpful.**
> The experiment in section 4.1 on “class-specific numbering” is a toy experiment where the dataset has equal elements in the input set. Set-equivariant functions cannot make differing predictions for equal elements, which results in 0% accuracy as reported in Table 2. We will mention this in the paragraph.
>
> **The figures should be ordered correctly and stretch across too many pages. Currently, the text talks about, in order, the results in figure 1, table 2, and table 3. However, the actual figures appear in the order of table 2, figure 1, table 3. Furthermore, the figures do not necessarily all need to be at the top of a new page, especially as table 2 pushes too far away from its source text.**
> We will improve the organization of the figures and tables.
>
> **The paper should include a related work section discussing more implicit differentiation works. In addition to classical deep implicit layers works like [1, 2], I also recommend implicit layers work on other non-Euclidean domains beyond sets such as [3, 4].**
> Thanks, we were not aware of these particular references on implicit layers for non-Euclidean domains before. We will add this to the manuscript.
>
> **The difference between the two "Slot Attention" rows in table 3 is not made clear.**
> The difference is mentioned in the caption of table 3, and we will make sure to make this clearer. The last paragraph in Appendix D.3 describes this in more detail already, but we appreciate that it is currently somewhat buried with a lot of other details.

---

> > ### Comment · Reviewer_KAp5 · 2021-11-30
> > **Score Update**
> >
> > Thank you for addressing many of my concerns. I am moving my score up to a 6.

---

### Official Review · Reviewer_4sjy · 2021-11-02

**Correctness:** 4
**Technical Novelty And Significance:** 3
**Empirical Novelty And Significance:** 4
**Recommendation:** 8
**Confidence:** 4

**Main Review:**

# Strengths
- The paper is well-written and understandable.
- Implicit DSPN leads to very strong performance and is a good application of the recent progresses in deep learning with implicit differentiation.
- The identification of the conceptual advantages of exclusive multiset-equivariance could serve as guidance towards which models future research will focus on. However, some issues have to be resolved, see weaknesses.

# Weaknesses & Open Questions
- Questions regarding importance of exclusive multiset-equivariance:
Is exclusive multiset-equivariance really a deciding property in practice? The authors make the following two points:
1) Set-equivariant networks cannot separate duplicate elements. This raises the question how often this happens in practice. The authors mention smoothing in graph problems (this is probably unlikely to produce exact duplicates), duplicates in the data (this is clearly the case in the toy experiment by design, but not obviously the case for the CLEVR benchmark), zeros from relu activations (to the best of my knowledge relu is not often used as the final nonlinearity) and duplicates from projections. It would be interesting to see how often duplicate elements actually show up, e.g. by reporting the percentage of instances for which this happens during the forward pass of a trained model on the CLEVR benchmark.
2) ‘set-equivariance […] places a limitation on [the set prediction models] modeling capacity even if exact duplicates are never encountered’. I agree that very similar elements will be a problem for many set-equivariant networks that try to approximate functions such as push-apart, however, there are also set-equivariant networks for which this is not the case. An example would be to employ a sorting operation that instead of randomly breaking ties outputs the mean of the duplicate elements multiple times, which I believe makes the jacobian set-equivariant but is still able to separate arbitrarily similar (non-equal) elements. A version of DSPN/iDSPN with this modified sorting operation (possibly with noise added to inputs to break exact ties) would be an interesting comparison in the experiments as well, as it is the closest related architecture that breaks exclusive multiset-equivariance. If (even with added noise) this modified architecture fails to achieve similar performance this would really show the advantage of the exclusive multiset-equivariant architecture.

- While producing impressive results, the technical contribution of iDSPN in itself is not very strong, as it is a straightforward application of implicit differentiation to DSPN.

**Summary Of The Paper:**

This paper the discusses the conceptual improvements of exclusive multiset-equivariance over set-equivariance in the context of set prediction networks. Deep Set Prediction Networks (DSPN) are identified as the only used architectures for set prediction that satisfy exclusive multiset-equivariance with a specific choice of encoder that employs sorting. As an answer to DSPN not being the leading architecture in terms of performance on set prediction benchmarks, iDSPN is proposed as an improved version of DSPN, which still satisfies exclusive multiset-equivariance and builds on implicit differentiation to employ better optimizers and reduce memory usage and computation time. The experiments highlight the benefits of exclusive multiset-equivariance in toy tasks and demonstrate that iDSPN significantly outperforms the state-of-the-art on the CLEVR object property prediction benchmark.

**Summary Of The Review:**

The conceptual advantage of exclusive multiset-equivariance is interesting but lacks some additional comparisons in order to demonstrate that it is the deciding property in practice.
Implicit DSPN works very well but on its own is just an application of existing methods to a new task, which is not necessarily bad but limits the technical novelty of this part of the contribution.

---

> ### Author Response · Authors · 2021-11-11
> **Response to reviewer 4sjy**
>
> **This raises the question how often this happens in practice**
> In general, we don’t expect exact duplicates to appear frequently in practice. An important point is that while multiset-equivariance is defined in terms of exact duplicates, the behavior that is most relevant in practice is for *similar elements*, not exact duplicates. For a continuous set-equivariant model (i.e. most other set predictors like slot attention, set transformers, equivariant Deep Sets, etc.), impossibility to separate equal elements means difficulty to separate similar elements. We plan to adjust the introduction to state this more clearly.
>
> **An example would be to employ a sorting operation that instead of randomly breaking ties outputs the mean of the duplicate elements multiple times**
> This is an excellent point. With the suggested operation, arbitrarily similar but non-equal elements can be separated due to the discontinuity provided by the sorting operation, and arbitrarily small noise handles exact equals in the case of experiment 1. We fully agree that this causes problems for the current definitions and in experiments, we expect the proposed sorting variant to lead to virtually the same results as using normal sorting.
>
> We are working on fixing the definition by making the distinction in set-equivariance between the function $f$ being continuous or discontinuous. Most of the discussion in the paper should remain the same (since the set-equivariant models in the literature are all continuous). The definition for multiset-equivariance will stay the same too, we just have this additional case of a discontinuous set-equivariant function that the reviewer proposed that can’t separate equals (without noise) but can separate arbitrarily similar elements. This change does not significantly affect the overall message of the paper, but it’s an important change for correctness. We highly appreciate the reviewer for spotting this subtle point.
>
>
> **lacks some additional comparisons in order to demonstrate that it is the deciding property in practice**
> We are adding additional experimental results comparing an exclusively multiset-equivariant iDSPN (through FSPool) to a continuous set-equivariant iDSPN (through sum/mean pooling), please see our response to reviewer PV3V for the details on this.

---

> > ### Comment · Reviewer_4sjy · 2021-11-28
> > **Score update**
> >
> > I acknowledge the significant changes made by the authors in the updated version of the paper, and will raise my score to an (8).

---

### Official Review · Reviewer_PV3V · 2021-11-03

**Correctness:** 2
**Technical Novelty And Significance:** 3
**Empirical Novelty And Significance:** 3
**Recommendation:** 5
**Confidence:** 3

**Main Review:**

The authors consider an interesting problem in the construction of neural network models designed to be applied on set objects. The contribution of the authors is two-fold: 1) the authors formalize a discussion on multiset-equivariance and the limits of set-equivariance, and 2) the authors propose an improvement upon deep set prediction networks through the use of implicit gradients. In general, I found the treatment of (1) to be interesting, but somewhat inadequate as there appears to be remaining mathematical issues (see details below). Additionally, I was not convinced that multiset-equivariance is of practical importance in general problems of interest. On the other hand, I found the treatment of (2) to be a valuable contribution, with promising empirical results.

**On the treatment of multiset-equivariance**
I was confused by some claims on multiset-equivariant (but non-set-equivariant) functions made in the paper, as not all the constructions used in the paper appear well-defined. For example, the construction in Appendix A (Jacobian of sorting) appears particularly ill-defined, as the authors refer to “the permutation matrix”, which is not unique whenever there are equal elements. One possibility here would be to e.g. consider a stable sort, which breaks ties in the sequential order of the input, but this is a function on tuples and not sets, and I believe would be set-equivariant in any case.

I was also uncertain about the treatment of element labels in the context of multiset-equivariant functions where equal elements may take different values. In particular, when we have labels attached to each input element, allowing non set-equivariant functions then would require a further permutation invariant matching step to reconcile labels of equal elements (as the outputs for equal elements are produced in arbitrary order). Given this, it would be great if the authors could address how this problem is tackled in their applications to counting equal elements in a multiset, and also provide a generic framework for tackling this important issue in using non set-equivariant functions.

**On the empirical importance of multiset-equivariance**
It is not clear to me that typical set prediction problems are particularly constrained by the set-equivariance assumption used in the model. Indeed, as the authors note, slot attention (which is set-equivariant) performs better than DSPN on some of the tasks at hand. Perhaps a comparison of a set-equivariant and a strictly multiset-equivariant iDSPN architecture on a variety of tasks could help demonstrate the importance of being able to assign different values to identical inputs.

**On the empirical results**
I generally found the empirical results promising, although I believe it could be improved by including some of the following. 1) In table 2, a natural question would be if the apparent gap in required sample size could be mitigated or closed through data augmentation for non-equivariant models. 2) In table 3, it would be helpful to also include a like-for-like comparison of the iDSPN model with the same number of iterations as the DSPN model, to assess the potential degradation due to inaccurate gradient, and also compare the speed-up in a like-for-like scenario.


**Summary Of The Paper:**

The authors examine the problem of building multiset-equivariant neural networks, and note that there exists multiset-equivariant functions which are not set-equivariant. Given this observation, using set-equivariant models for multiset prediction tasks may be limiting. The authors propose a strategy to construct multiset-equivariant functions (which are not set-equivariant) through the Deep set prediction network framework. The authors propose to improve upon that framework by making use of implicit differentiation to compute a backwards mode gradient without backpropagating through the optimization process which is memory- and compute- intensive. Some empirical results are provided for a simulated toy dataset, as well as a simulated object property prediction problem.


**Summary Of The Review:**

The authors tackle an intriguing problem in modelling functions on multi-sets. The paper is overall promising, and I would be in favor of accepting if the authors could clarify some issues on the theory and add some more details in the empirical results.

---

> ### Author Response · Authors · 2021-11-11
> **Response to reviewer PV3V 3/3**
>
> **On the empirical results**
> We are performing the two suggested experiments as well and will add them to the paper once they are complete.
>
> Experiment in Table 2:
> Since we train these models for a sufficient number of steps, including data augmentation (random permutation) for the not-equivariant models makes the 1x setting essentially the same as 100x with accuracies > 95% for BiLSTM and Transformer with PE, matching iDSPN results. When we increase the number of possible classes from 4 to 8, the not-equivariant models see a smaller fraction of the possible permutations for a multiset even with data augmentation, so their accuracy drops for smaller sample sizes: at 1x samples, BiLSTM gets 62%, Transformer with PE 44%, iDSPN 91%. At 100x samples, Transformer with PE and iDSPN are above 95% again, only BiLSTM is worse than that with ~66%.
>
> A completely like-to-like comparison between iDSPN and DSPN on CLEVR is not entirely possible: there are optimization issues when trying to use the exact same hyperparameters in DSPN, we have to reduce the momentum in the inner optimization to make it work at all. After reducing the momentum in DSPN+FSPool from 0.9 to 0.5, the AP_0.125 is only 40%, which is *worse* than iDSPN+FSPool at 79%. iDSPN appears to be more stable to the specific inner optimization procedure, so in the setting as close to like-to-like as we can get for now, iDSPN is actually doing better than DSPN. So, it’s better to think of iDSPN and DSPN as separate procedures, rather than simply as iDSPN approximating the gradient of DSPN suggesting that iDSPN can’t be better than DSPN. We will re-run iDSPN+FSPool with this reduced momentum for a better like-to-like comparison as well.
> For the speed-up of iDSPN over DSPN: training takes 2 h 40 min for iDSPN and about 3 h 15 min for DSPN. Much of the computation is spent in the ResNet18 image encoder.

---

> ### Author Response · Authors · 2021-11-11
> **Response to reviewer PV3V 2/3**
>
> **Perhaps a comparison of a set-equivariant and a strictly multiset-equivariant iDSPN architecture on a variety of tasks could help demonstrate the importance of being able to assign different values to identical inputs**
> We are now running additional experiments with iDSPN using sum/mean pooling (which make it set-equivariant) on the second and third experiment to try to better show the importance of exclusive multiset-equivariance.
>
> - There is no need to run the first experiment (section 4.1) with a set-equivariant DSPN since we know that the task is impossible for set-equivariant models, it will just give us 0% accuracy.
> - Experiment 2 (random set autoencoding) with iDSPN+Sum results in much worse reconstruction losses than iDSPN+FSPool. Here are some examples, averages over 8 runs: on n=2 d=32, iDSPN+Sum 40 iterations is worse in reconstruction loss than iDSPN+FSPool (all else equal) by a factor of ~10. On n=4, d=4, it is worse by a factor of ~10. On n=8, d=16, it is worse by a factor of ~21. These results are not cherry-picked: every dataset combination we checked, iDSPN+Sum was worse by around this much. For n=16 and n=32 it almost always diverged, so we are currently waiting for iDSPN+Mean results for those (preliminary results for smaller n are fairly similar to iDSPN+Sum). Keep in mind that as we explain in section 4.2, these dataset configurations can be considered to be more challenging than many real-world set autoencoding/prediction tasks due the lack of global structure for the model to use as a shortcut, so we believe that these results are reasonably representative for real-world tasks as well.
> - For experiment 3 (CLEVR), the AP_0.125 is about 47% for iDSPN+Sum as compared to 79% for iDSPN+FSPool, so there is still quite a difference between them, though not as large as between iDSPN+FSPool and Slot Attention. iDSPN+Mean is worse here with only 32% AP_0.125.
>
> These (preliminary) experiments show that the exclusively multiset-equivariant iDSPN+FSPool seem to be consistently better than the set-equivariant iDSPN+Sum. When these experiments are fully completed, we will incorporate them into the paper.
>
> For another point on the importance of being able to assign different values to identical inputs:
> One difference between set-equivariant and exclusively multiset-equivariant models we did not mention in the paper is the initialization $Y_0$. For example, Slot Attention and TSPN randomly sample the elements for every input with a learnable variance, but the model has to strike a balance between too low variance (elements are too similar, difficult to separate due to set-equivariance) and too high variance (too much variety, difficult for the model to learn to be accurate for all initializations across a wide value range).
> When using an exclusively multiset-equivariant model, setting every entry in $Y_0$ to be zero is perfectly fine. Exclusive multiset-equivariance means that there is no problem separating the all-0s elements. This avoids the potential issue of having to balance the variance. We observed no difference in results on CLEVR in earlier tests when using all-0s initialization. In the final experiments in the paper, we used a random initialization with a fixed small variance only because we wanted to make iDSPN results better comparable to the Slot Attention setup for that experiment.

---

> ### Author Response · Authors · 2021-11-11
> **Response to reviewer PV3V 1/3**
>
> **the construction in Appendix A (Jacobian of sorting) appears particularly ill-defined, as the authors refer to “the permutation matrix”**
> With “the permutation matrix”, we simply refer to whatever permutation matrix is used by a specific sorting algorithm. There can be multiple permutation matrices that would result in the same sorted result, but a sorting algorithm will have to choose one of these matrices. Regardless of which of the equivalent permutations is chosen, that permutation is the $S_X$ we refer to and it will be exclusively multiset-equivariant as we prove in that appendix. The aspect of stable vs unstable sort does not matter here, it is exclusively multiset-equivariant either way.
> You will notice that push_apart also uses the position in its input to break the tie. Using the position is not a problem because as we mention in the introduction and section 2.1, models work with the list/tuple representation. Some operations that work with lists/tuples simply satisfy additional properties (i.e. permutation-invariance, set-/multiset-equivariance) that makes them useful in a multiset context too. There is nothing inherently wrong with making use of the position in the list representation as long as we still get the properties we want. In this case, making use of it to determine the permutation is not a problem for sorting (it’s permutation-invariant) nor for its Jacobian (it’s multiset-equivariant), so this is perfectly fine.
>
> We appreciate that this wasn’t entirely clear from appendix A and we will update it to clarify this. Let us know if that makes things clearer and if there are any other mathematical issues you had in mind.
>
> **when we have labels attached to each input element, allowing non set-equivariant functions then would require a further permutation invariant matching step to reconcile labels of equal elements (as the outputs for equal elements are produced in arbitrary order).**
> You are correct about the matching, we describe the experimental setup for this in Appendix D.1 in more detail. All the models (equivariant or not) in section 4.1 use a setup where Hungarian matching is applied on the output sets to match it to the ground-truth. Basically, we perform a Hungarian matching within each class. A simple way to implement this class-specific matching with a single call to the matching algorithm is to increase the pairwise cost for matching two elements of different classes. Other than that, the matching is exactly the same as is used in normal set prediction like in the random sets and CLEVR experiments. We can expand the description of this in Appendix D.1 if desired.
>
> **Indeed, as the authors note, slot attention (which is set-equivariant) performs better than DSPN on some of the tasks at hand.**
> Our argument is that DSPN *should* perform better than slot attention due to exclusive multiset-equivariance being less restrictive than slot attention, but because of other limitations in DSPN this was not the case before. There are of course more factors than just the single property of exclusive multiset-equivariance that determine the performance of a model. We hypothesized that DSPN should have an advantage due to exclusive multiset-equivariance, but suffers from insufficient minimization. So, we improved this aspect of DSPN by introducing the implicit differentiation version and we show that indeed, the new iDSPN handily improves over Slot Attention. So, iDSPN (which we argued should perform better than slot attention due to exclusive multiset-equivariance) performs better after all.

---

### Author Response · Authors · 2021-11-11
**Response to all reviewers**

We thank all the reviewers for their high-quality reviews and the many valid points on areas of improvement. We are now working on making the requested changes and clarifying things. We also encourage all the reviewers to check our response to reviewer PV3V (part 2 and part 3) since it contains some (preliminary) results on additional experiments that strengthen the importance of multiset-equivariance in section 4.2 and 4.3. We will add the additional experiments to the paper once complete.

---

### Author Response · Authors · 2021-11-18
**Updates to the paper**

We have made an update to the paper to address the reviewers’ comments. This revision should address all the points the reviewers have made and clarify possible misunderstandings, and we hope that the reviewers will reconsider their scores. Please let us know of anything else that could be improved or clarified. We summarize the changes below and mark which reviewer comment each change addresses.

## Requested experiments to strengthen results

We added the requested experiments to Appendix E (which previously contained extended experimental results only for Section 4.2), which strengthens our existing results. The additional evidence we provide for the benefits of exclusive multiset-equivariance and implicit DSPN should hopefully address the concerns of the reviewers about the generality of the results. We believe that this addresses the main concern of the reviewers.

Specifically, the new experiments aim to evaluate:
- **[PV3V, 4sjy, MvMj]** exclusive multiset-equivariance vs (continuous) set-equivariance for the experiments in Section 4.2 and 4.3 (Appendix E.2, Appendix E.3). The initial submission only contained such experiments in 4.1. We compare iDSPN using an FSPool encoder (exclusively multiset-equivariant) to iDSPN with Sum or Mean pooling encoder (set-equivariant). These results provide additional evidence of the benefit of exclusive multiset-equivariance over set-equivariance, especially for the random sets autoencoding. With all else equal, the exclusively multiset-equivariant iDSPN consistently improves over the set-equivariant iDSPNs in every setting, and for random sets autoencoding often with a loss that is lower by an order of magnitude.
- **[PV3V]** iDSPN vs DSPN in Section 4.3 in a like-to-like comparison (Appendix E.3). In the same setup, iDSPN results are actually better than DSPN results and appear to be more stable for different choices of the momentum parameter in the inner optimization. iDSPN is of course faster and more memory-efficient than DSPN as well.
- **[PV3V]** The effect of data augmentation in Section 4.1 for the not-equivariant models (Appendix E.1). This shows that while in the existing setting (set size 64, 4 possible classes) data augmentation would restore the performance of these baselines fully, data augmentation doesn’t help as much in the slightly more difficult setting with 8 instead of 4 possible classes.
In Section 4, we add short summaries with cross-references to each of these new results as part of the main text.


## New analysis to address a theoretical problem

- **[4sjy]** To address the issue with the definition that 4sjy pointed out through their augmented sorting example, we add the new Appendix B that discusses *continuity on multisets* in detail, why it’s desirable, and its relation to the equivariance properties we have been talking about. The main discussion in Section 2 remains the same since it is mostly about properties for exact equals. Continuity on multisets lets us talk about what happens for similar elements, which addresses the augmented sorting example the reviewer raised. In particular, we show that the Jacobian of sorting operation used in iDSPN is multiset-continuous and exclusively multiset-equivariant, which we prove is a *sufficient condition* for being able to separate similar elements. The augmented sort is multiset-discontinuous and set-equivariant: this combination of properties is not sufficient for separating similar elements in general.
With our new analysis of multiset-continuity, we also make the discussion on open problems in section 6 more precise.

## Improvements to clarity to reduce misunderstandings

- **[4sjy]** We make it clearer in the introduction and the “The problem with set-equivariance” paragraph that the main problem is with similar elements, not necessarily exact equal elements. The example reasons in the paragraph are now about why elements might be/become similar (not necessarily equal).
- **[KAp5]** We clarify in  “The problem with set-equivariance” that experiment 4.1 showcases the problem with set-equivariance directly and how a function like push_apart can be relevant to a real-world use case for point clouds.
- **[PV3V]** In Section 4.1, we state that a matching loss is used for all models, with a cross-reference to Appendix F, where we added some detail on how this works.
- **[KAp5, MvMj]** In Section 4.3, we now specify that the explanation of Slot Attention* and Slot Attention$\dagger$ can be found in the table caption.
- **[PV3V]** We explain better why the sorting is not ill-defined. In Appendix A (The Jacobian of sorting is exclusively multiset-equivariant) we elaborate what happens for $S_X$ when there are multiple valid permutations that would lead to a correct sorting.
- **[MvMj]** In Appendix D (Derivation of Implicit DSPN) we explain the notation we use for the gradient of $L$ evaluated at a point a bit more, which we believe was the point of confusion.

---

> ### Author Response · Authors · 2021-11-18
> **Updates to the paper**
>
> ## Extended related work
> - **[KAp5]** In Appendix C (Additional related work) we include a discussion on related implicit differentiation work.
>
> ## Changes to structure
> - **[KAp5]** We moved the tables and figures around a bit to be closer to the relevant text.
> - We moved the last paragraph in the related works (on relating multiset-equivariance to the multiset-invariance in the literature) into the extended related works in Appendix C to save some space.

---

### Decision · Program_Chairs · 2022-01-20

**Decision:**

Accept (Poster)

**Comment:**

The paper points out how set equivariant functions limit the types of functions that can be represented on multisets. They develop an new notion of multiset equivariance to address this limitation. The paper improves an existing multi-set equivariant Deep Set Prediction Network through implicit differentiation, which is an area of rising interest. The reviewers and I note that the paper is well written.